



# Measurements of hydroperoxy radicals (HO$_2$) at atmospheric concentrations using bromide chemical ionization mass spectrometry

Sascha R. Albrecht[1], Anna Novelli[1], Andreas Hofzumahaus[1], Sungah Kang[1], Yare Baker[1], Thomas Mentel[1], Andreas Wahner[1], and Hendrik Fuchs[1]

[1]Forschungszentrum Jülich GmbH, Institute of Energy and Climate Research, Troposphere (IEK-8), 52428 Jülich, Germany

**Correspondence:** Sascha Albrecht (s.albrecht@fz-juelich.de)

**Abstract.**

Hydroxyl and hydroperoxy radicals are key species for the understanding of atmospheric oxidation processes. Their measurement is challenging due to their high reactivity, therefore very sensitive detection methods are needed. Within this study, the measurement of hydroperoxy radicals (HO$_2$) using chemical ionization combined with an high resolution time of flight

mass spectrometer (Aerodyne Research Inc.) employing bromide as primary ion is presented. The $1\sigma$ limit of detection of $4.5 \times 10^7 \, \mathrm{molecules \, cm^{-3}}$ for a $60\,\mathrm{s}$ measurement is below typical HO$_2$ concentrations found in the atmosphere. The detection sensitivity of the instrument is affected by the presence of water vapor. Therefore, a water vapor dependent calibration factor that decreases approximately by a factor of 2 if the water vapor mixing ratio increases from 0.1 to 1.0 % needs to be applied. An instrumental background most likely generated by the ion source that is equivalent to a HO$_2$ concentration of

$1.5 \pm 0.2 \times 10^8 \, \mathrm{molecules \, cm^{-3}}$ is subtracted to derive atmospheric HO$_2$ concentrations. This background can be determined by overflowing the inlet with zero air. Several experiments were performed in the atmospheric simulation chamber SAPHIR at the Forschungszentrum Jülich to test the instrument performance by comparison to the well-established laser-induced fluorescence (LIF) technique for measurements of HO$_2$. A high linear correlation coefficient of $R^2 = 0.87$ is achieved. The slope of the linear regression of 1.07 demonstrates the good absolute agreement of both measurements. Chemical conditions during

experiments allowed testing the instrument's behavior in the presence of atmospheric concentrations of H$_2$O, NO$_\mathrm{x}$ and O$_3$. No significant interferences from these species were observed. All these facts are demonstrating a reliable measurement of HO$_2$ by the chemical ionization mass spectrometer presented.

# 1 Introduction

Understanding of the oxidation processes in the atmosphere requires sensitive measurements of the radical species involved. Hydroxyl radicals (OH) are the most important oxidative species and are highly reactive to most of the inorganic and organic





pollutants in the atmosphere. Primary sources of OH radicals are mainly ozone photolysis and in polluted environments also nitrous acid (HONO) photolysis can be of importance. Organic pollutants are oxidized by OH to produce organic peroxy radical species ($RO_2$) and also hydroperoxy radicals ($HO_2$). OH and $HO_2$ radicals are closely inter-connected by a radical chain reaction, in which OH is reformed by the reaction of $HO_2$ with nitric oxide (NO):

$HO_2 + NO \rightarrow OH + NO_2$ (R1)

As the atmospheric lifetime of $HO_2$ radicals is typically up to a factor 10 longer than that of OH radicals, $HO_2$ can be regarded as an important chemical reservoir for hydroxyl radical (OH). Atmospheric NO concentrations are often sufficiently high to maintain an efficient OH production by the reaction of $HO_2$ with NO, so that R1 provides a large portion of the total OH production. Measurements of both species are needed to analyze the OH radicals budget.

The majority of the techniques currently applied to measure atmospheric concentrations of $HO_2$ radicals use chemical conversion, which is an indirect measurement. In chemical amplifying systems, a radical reaction cycle between OH and $HO_2$ is established by adding two reactants. The concentration of the product species is therefore amplified compared to the small, initial $HO_2$ concentration in the sampled air.

PEroxy RadiCal Amplification (PERCA) instruments make use of NO and CO for the conversion of $HO_2$ to OH and OH to $HO_2$, respectively. One $NO_2$ molecule is produced in each reaction cycle so that the initially small $HO_2$ concentration is
amplified as $NO_2$, which is then detected by a luminol detector, fluorescence or absorption methods. Because $RO_2$ is also converted to $HO_2$ in the reaction with NO, these instruments measure the sum of $RO_2$ and $HO_2$. Typically an amplification of roughly a factor of 100 is achieved to produce a measurable amount of $NO_2$ (Cantrell et al., 1984; Hastie et al., 1991; Clemitshaw et al., 1997; Burkert et al., 2001; Sadanaga et al., 2004; Mihele and Hastie, 2000; Green et al., 2006; Andrés-
Hernández et al., 2010).

Alternatively to CO, $SO_2$ can be used in the chemical amplifier system (Reiner et al., 1997; Hanke et al., 2002; Edwards et al., 2003; Hornbrook et al., 2011). The high sensitivity of CIMS measurement using $NO_3^-$ as primary ion allows to detect $H_2SO_4$ produced in the reaction of $SO_2$ with OH. Amplification factors of approximately 10 are sufficient is this case. Like in the PERCA instrument, $RO_2$ is also converted to $HO_2$ in the reaction with NO in these instruments. However, Hornbrook et al.
(2011) developed a method to distinguish between $HO_2$ and $RO_2$ by operating the instrument at different chemical conditions (varying NO, $SO_2$ and $O_2$ concentrations), thereby changing the relative sensitivities for $HO_2$ and $RO_2$.

Laser-induced fluorescence (LIF) is a sensitive technique for OH radical measurements and it is used for the indirect detection of $HO_2$ by its conversion into OH after reaction with NO. The concurrent conversion of some specific $RO_2$ radicals can contribute to the $HO_2$ signal (Fuchs et al., 2011; Whalley et al., 2013; Lew et al., 2018). This can be minimized by reducing
the NO concentration added to the sampled air for the conversion of $HO_2$ to OH, but on the cost of a reduced sensitivity. A comparison of three LIF instruments in 2010 before the $RO_2$ interference was discovered showed significant differences in measured $HO_2$ concentration in experiments in the SAPHIR chamber (Fuchs et al., 2010). This could have been partly due to interferences from $RO_2$, but measurements also differed depending on the water vapor concentration.





Several drawbacks are connected with existing $HO_2$ detection methods. The PERCA systems exhibit a strong water vapor dependence of the amplification factor. In addition, chemical conversion of $HO_2$ by the reaction with NO used in all instruments can lead to the concurrent conversion of $RO_2$.

Previous work by Veres et al. (2015) showed that $HO_2$ radicals can be detected with a CIMS instrument using iodide as
primary ion. Sanchez et al. (2016) demonstrated for the first time that this approach can also be used with $Br^-$. The $HO_2$ radicals are directly measured by a mass spectrometer as an ion cluster formed with bromide ions. In this study, the direct measurement of atmospheric concentrations of $HO_2$ radicals using Br-CIMS is presented. A detailed characterization of the instrument has been performed. Further, the inter-comparison with an LIF based $HO_2$ measurement is used to identify potential interferences.

## 2    Methods

### 2.1    Chemical ionization mass spectrometry technique

The instrument used for the detection of the $Br^- \cdot HO_2$ cluster consists of a custom-built ion flow tube (Fig. 1) that is mounted upstream of a commercial, high resolution time-of-flight mass spectrometer (TOF-MS, Aerodyne Res.). For the detection of reactive $HO_2$ radicals, losses in inlets can play a significant role. As radical species are easily lost by contact on walls, the
inlet of the instrument is designed to sample air directly into the ion flow tube without additional inlet lines. The TOF-MS is equipped with an atmospheric pressure ionization (APi) transfer stage providing the ion transfer from the ion flow tube to the detector. The TOF mass analyzer (Tofwerk Ag, Switzerland) has a mass resolution better than 2000.

Ambient air containing $HO_2$ (flow rate 3.4 slm) is sampled through a 0.7 mm skimmer nozzle and is mixed with the bromide ions in the ion flow tube shown in Fig. 1. The ion flow tube has an inner diameter of 22 mm and a length of 130 mm. The
distance between the ion source and the nozzle downstream is 100 mm. The ion flow tube is kept at a constant pressure of 120 hPa using a butterfly control valve upstream of a scroll pump. Assuming that 5.4 slm of gas are passing through the ion flow tube without considering the complex fluid dynamics in the ion flow tube, the mean residence time is 4 ms. Longer versions of the ion flow tube of up to twice its size were tested, but a reduced sensitivity for $HO_2$ was found. Downstream of the ion flow tube, the sampled air enters a commercially available transfer stage (CI-API transfer stage, Aerodyne Research
Inc.) through a nozzle with 0.5 mm diameter. The transfer stage consists of two quadrupoles and direct current transfer optics that guide the ions to the TOF analyzer.

Bromide ions easily clusters with polar species e.g. acids (Caldwell et al., 1989). This enables their detection in the gas phase including $HO_2$, which is a relative strong acid (the binding energy is $353 \, kcal \, mol^{-1}$ Harrison (1992)). In order to produce $Br^-$ ions, a gas flow of 2 slm nitrogen is mixed with 10 sccm of a 0.4 % mixture of $CF_3Br$ in nitrogen (Air Liquide Deutschland
GmbH, $N_2$ 99.9999 % purity). The resulting gas mixture of approximately 20 ppmv $CF_3Br$ in nitrogen is supplied to the 370 MBq $^{210}Po$ ion source to generate bromide ions.

The isotopic pattern of bromide (approx. 1 $^{79}Br$ : 1 $^{81}Br$) provides additional information if a signal detected at a certain mass contains a cluster with bromide, because similar signals need to be contained at two masses (m/z and m/z+2). Therefore,


$HO_2 \cdot Br^-$ is detected on masses 112 and 114 with similar intensities. Both signals can be used for the data evaluation in order to improve the signal-to-noise ratio.

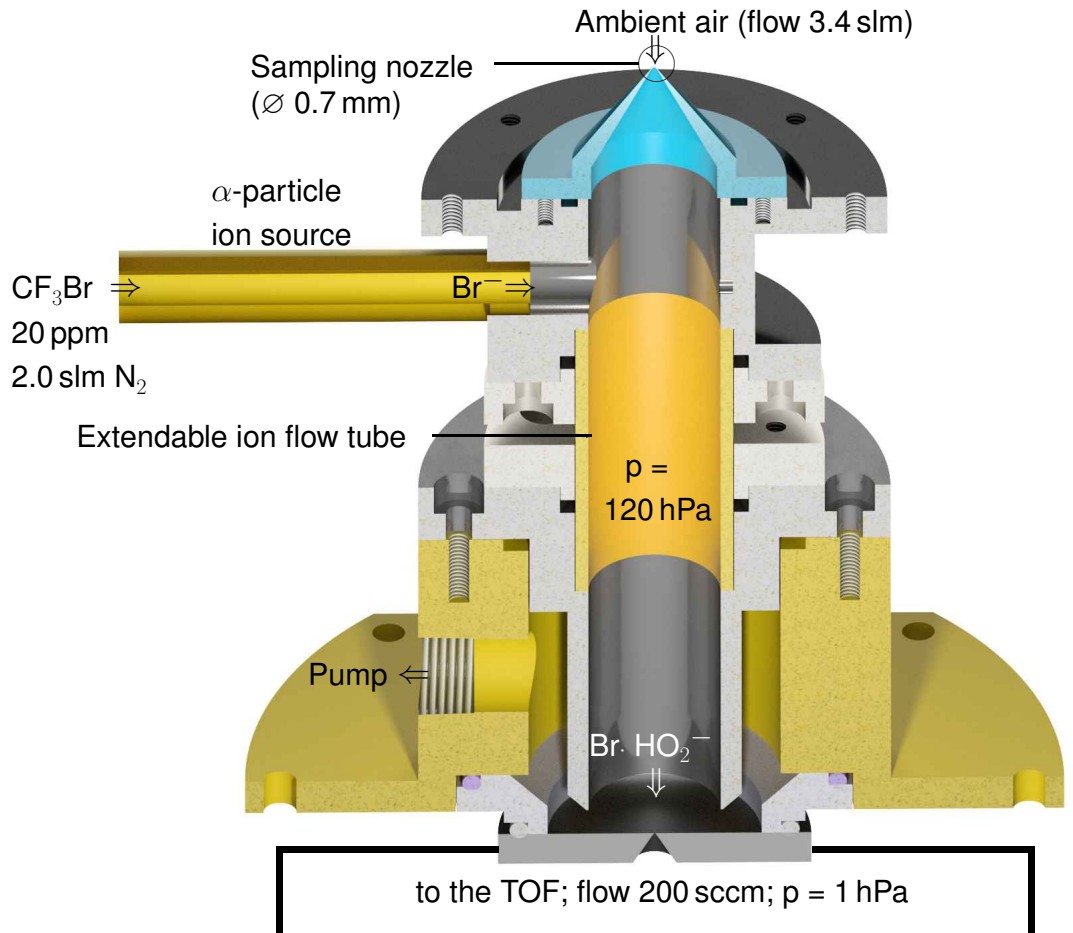

**Figure 1.** Schematic drawing of the ion flow tube, where $HO_2$ clusters with $Br^-$ are formed. The ion flow-tube is mounted upstream of an Aerodyne Time-of-Flight mass spectrometer.

The data are analyzed using the following procedure. 30 mass spectra measured with a time resolution of $2\,s$ are summed up to improve the signal-to-noise ratio (cf. Sect. 3.2). The $HO_2 \cdot Br^-$ ion cluster ion count rate (m/z 112) is normalized to
5   the count rate of the primary ion (m/z 79). The isotopic signal at a mass-to-charge ratio of 114 and 81 are treated in the same way. The signal at both isotopic masses of the $HO_2 \cdot Br^-$ ion cluster are compared to check for possible interference from ions not containing a bromide molecule. In the following step, a water vapor dependent sensitivity is applied to convert the signal to a $HO_2$ concentration. Details about the water vapor dependent sensitivity are presented in Sect. 3.1. Finally a constant background is subtracted from the data. No difference in the isotopic signals was observed showing that no other molecule





(not containing bromide) is interfering. In this study, only data from one of the two isotopes (m/z 112 and 79) are discussed for simplicity.

## 2.2 Calibration source

For calibrating the $HO_2$-CIMS instrument's sensitivity the same radical source is used as for calibration of the LIF instrument
that is in operation at Forschungszentrum Jülich (Fuchs et al., 2011). This is possible because the designs of the inlet nozzle and flow rates of both instruments are similar. The LIF is sampling 1.0 slm and the CIMS instrument is sampling 3.4 slm. Both flows are much smaller than the total flow through the calibration source. The calibration source provides a laminar gas stream of humidified synthetic air at a flow rate of 20 slm. The gas supply device for the calibration source allows for systematic variation of the water vapor concentration. During calibrations, the water vapor concentration is altered from 0.1 to
1.6 %, in order to determine the humidity dependence of the instrument's sensitivity. Water vapor is photolysed at 185 nm at atmospheric pressure using a penray lamp leading to the production of equal concentrations of OH and $HO_2$ radicals (Fuchs et al., 2011). The radical concentration that is provided by the calibration source is calculated from the UV intensity that is monitored by a photo-tube detector, the flow rate and water vapor concentration. The photo-tube signal is calibrated against ozone that is concurrently produced from oxygen photolysis by the 185 nm radiation. An absorption cell in-between the UV
lamp and the photolysis region can be filled with a $N_2O$ / $N_2$ mixture to vary the UV intensity, as $N_2O$ is a strong absorber at this wavelength. If excess CO is added to the synthetic air provided to the calibration source, OH is converted to $HO_2$, so that the $HO_2$ concentration is doubled compared to the operation without CO. Typically, the calibration is performed at $HO_2$ concentrations between $5 \times 10^8$ and $1 \times 10^{10}$ molecules cm$^{-3}$.

## 2.3 HO$_2$ detection by laser-induced fluorescence

The LIF instrument uses two detection channels to detect OH and $HO_2$ simultaneously. The LIF instrument has been described in detail by Holland et al. (2003), Fuchs et al. (2011), and Tan et al. (2017).

For the $HO_2$ measurement, a gas stream of ambient air is expanded in to the sample cell at 4 hPa. NO is added to the sampled air for the conversion of $HO_2$ to OH (Reaction R1). The NO concentration is adjusted to provide a $HO_2$ conversion efficiency of approximately 10 % in order to minimized concurrent $RO_2$ conversion (Fuchs et al., 2011). The OH radicals
are excited by a laser pulse at 308 nm, provided by a dye laser system. Ozone can be photolysed at 308 nm, which can lead to a small interference from ozone that is subtracted from the measured signal. For the experiments discussed here, 50 ppbv $O_3$ gave a signal that is equivalent to a $HO_2$ concentration of $3 \times 10^6$ cm$^{-3}$. The sensitivity of the $HO_2$ LIF detection is water vapor dependent due to the quenching of the OH fluorescence by water. The change in the sensitivity is calculated from quenching constants. Both corrections are taken into account. The accuracy of the LIF $HO_2$ measurement is $\pm 10$ % from the
uncertainty of the calibration. The typical precision of measurements gives an limit of detection of $1 \times 10^7$ mol cm$^{-3}$ ($2\sigma$) for a 80 s measurement (Tan et al., 2017).



## 2.4 SAPHIR

SAPHIR is an atmospheric simulation chamber at the Forschungszentrum Jülich. The chamber has been described in detail by Rohrer et al. (2005). It consists of a double-wall FEP film of cylindrical shape (length $18\,\mathrm{m}$, diameter $5\,\mathrm{m}$, volume $270\,\mathrm{m}^3$). It is equipped with a a shutter system that can be opened to expose the chamber air to natural sunlight. Synthetic air used in

the experiments is produced from liquid nitrogen and oxygen of highest purity (Linde, purity <99.9999 %). A combination of sensitive measurement instruments allows for studying chemical systems under well-defined, atmospheric conditions and trace gas concentrations. SAPHIR has proven to be a valuable tool for inter-comparison of different measurement techniques (Fuchs et al., 2012; Dorn et al., 2013; Fuchs et al., 2010; Apel et al., 2008), as it is ensured that all instruments can sample the same air composition.

For this study, measurements were performed during a series of experiments in the SAPHIR chamber in May and June 2017. The focus of the experiments was to study the chemistry of two classes of oxidation products of isoprene: the isoprene hydroxyhydroperoxides (ISOPOOH) and the isoprene epoxydiols (IEPOX). In addition, reference experiments without addition of VOCs, as well as experiments with isoprene were performed. These experiments were used to compare the performance of the CIMS and the LIF instrument at atmospheric $HO_2$ concentrations, testing various conditions, e.g. presence of ozone, $NO_x$

species and different water concentrations.

The CIMS was mounted at the bottom of the chamber, $4\,\mathrm{m}$ away from the LIF instrument. The ion flow tube setup shown in Fig. 1 was directly connected to the chamber, so that the sampling nozzle was sticking into the chamber.

Data from the following instruments are used for the data evaluation and interpretation: The humidity was measured using a Picarro cavity ring-down instrument (G2401 Analyzer). NO and $NO_2$ were monitored by a Eco Physics chemiluminescence

instrument (TR780) and ozone was detected by an UV photometer (41M, Ansyco).

## 3 Characterization of the $HO_2$-CIMS

### 3.1 Calibration procedure

In general, the conversion of ion count rates measured by a CIMS instrument to concentrations of the detected molecule requires regular calibrations of the sensitivity. For calibrating the $HO_2$ sensitivity, we utilized a radical source as described in

Sect. 2.2. Figure 2 shows the measured, normalized ion count rates measured by the CIMS, when the calibration source was operated at a constant water vapor mixing ratio of 1.0 %. The $HO_2$ concentration was varied by changing the UV radiation intensity, which was achieved by varying the $N_2O$ concentration in the absorption cell. A linear behavior for the normalized count rate is observed in a range of $3.0 \times 10^8$ to $1.3 \times 10^9$ $HO_2$ molecules $\mathrm{cm}^{-3}$. The slope of the linear regression gives the calibration factor of $6.8 \times 10^{-12}\,\mathrm{cm}^{-3}\,\mathrm{ncps}^{-1}$. The intercept of $5.1 \times 10^{-4}\,\mathrm{ncps}$ of the linear fit indicates a $HO_2$ background

signal. No background correction of the CIMS signal (see below) is applied here.

Alternatively, the $HO_2$ provided by the calibration source can be varied by changing the water mixing ratio at constant UV intensity. The $HO_2$ concentration provided by the calibration source is well characterized for different water mixing ratios. This





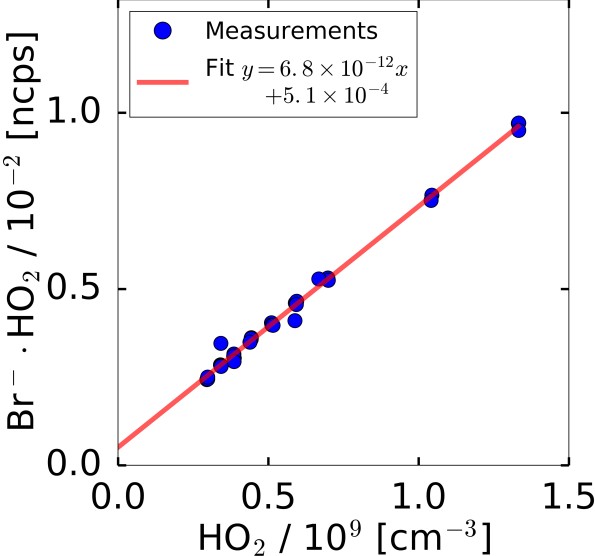

**Figure 2.** Count rate of $HO_2 \cdot Br^-$ ion cluster (m/z 112) normalized to the primary ion $Br^-$ (m/z 79) during sampling from the $HO_2$ calibration source. The $HO_2$ concentration provided by the source was varied by attenuating the radiation of the 185 nm radiation used to photolyse water. The water vapor mixing ratio was kept constant. The error bars are smaller than the symbols in the figure.

allows the determination of the water dependency of the CIMS. The water dependent sensitivity is defined by Eq. 1, where $c$ is the sensitivity that depends term on the water concentration.

$$[\text{HO}_2] = c(\text{H}_2\text{O}) \frac{m/z(112)}{m/z(79)} \tag{1}$$

Figure 3 shows the sensitivity determined for each water vapor mixing ratio showing a decreasing sensitivity with increasing
5   water vapor mixing ratio. The water dependent decrease in sensitivity is nearly linear for atmospheric relevant water mixing ratios higher than 0.1 %. Two effects contribute to the water dependence: The $HO_2$ ion cluster is stabilized by water during the attachment process, as water takes the access energy of the cluster rearrangement during substitution by the analyte molecule. On the other hand, the $HO_2$ bromide ion cluster is in a fast equilibrium with polar molecules in the gas phase. If atmospheric water vapor concentrations are present in the ion flow tube, water may substitute $HO_2$ in the ion cluster. The ion cluster
10   typically has a shell of water molecules at atmospheric conditions, caused by the ions polarity (Klee et al., 2014; Derpmann et al., 2012; Albrecht et al., 2014).

$$\text{HO}_2 + \text{Br}^- \cdot \text{H}_2\text{O} \rightleftharpoons \text{Br}^- \cdot \text{HO}_2 + \text{H}_2\text{O} \tag{R2}$$

As indicated in R2, an excess of water can push the reaction equilibrium in the reverse direction. Thereby, the cluster switching from $HO_2 \cdot Br^-$ to $H_2O \cdot Br^-$ causes a decrease in sensitivity.





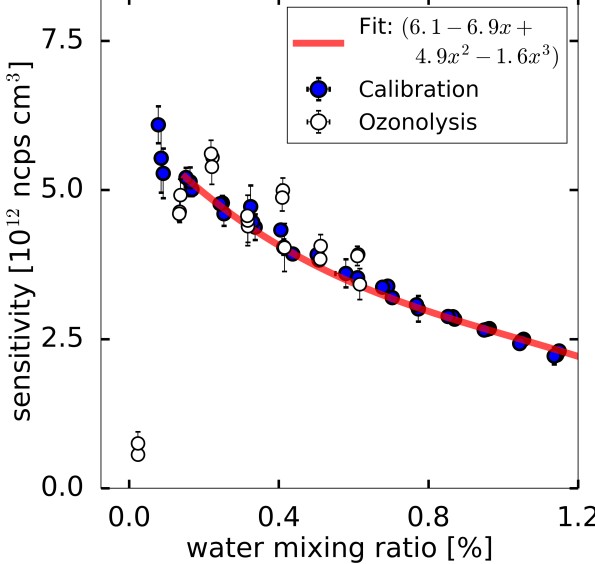

**Figure 3.** Measured $HO_2$ sensitivity as a function of the water mixing ratio in two experiments. For the calibration, $HO_2$ was produced by the radical source while varying the water vapor concentration which causes a change in the $HO_2$ radical concentration. During the ozonolysis experiment, $HO_2$ was produced from the ozonolysis of 2,3-dimethyl-2-butene, which is independent of the water vapor mixing ratio. The red line shows a third order polynomial fit applied to the calibration data.

Further, the $HO_2$ radical itself can form a water cluster (Aloisio and Francisco, 1998; Kanno et al., 2005; Stone and Rowley, 2005). This, for example, leads to an enhancement of the $HO_2$ self reaction of up to a factor of two for atmospheric conditions (Stone and Rowley, 2005). However, only a fraction of the $HO_2$ (20 % at 297 K and 50 % humidity) is attached to a water molecule at atmospheric conditions (Kanno et al., 2005). The concentration of $HO_2$ water radical clusters is further reduced

because of the lower pressure in the ion flow tube along with a lower partial water pressure. Therefore, compared to dry conditions an roughly 10x increased sensitivity at humid conditions is likely mainly caused by the ion water cluster.

A direct calibration for dry conditions was not possible with the radical source, because the calibration source needs water to generate $HO_2$. The sensitivity of the instrument was also characterized by the production of $HO_2$ from the ozonolysis of 2,3 dimethyl-2-butene, that was added in a concentration of 30 ppbv to a mix of synthetic air and 200 ppbv ozone. The radical

source was used as a flow-tube to overflow the inlet of the instrument with this gas mixture. 0.2 % CO was added to scavenge OH radicals produced from the ozonolysis reaction by a fast conversion of OH to $HO_2$. The water mixing ratio was altered during the ozonolysis experiment from 0.0 to 0.6 %. Assuming that the $HO_2$ concentration from the ozonlysis is constant, the relative change in the signal gives the relative change of the instrument sensitivity. In addition, calibration measurements using the water photolysis were performed for water vapor mixing rations higher than 0.1 %, so that the water dependence of the

sensitivity determined by the two methods can be compared. As shown in Fig. 3, the instrument response is similar in both experiments. In addition, the instrument's sensitivity at dry conditions could be tested showing that the instrument sensitivity




drops by nearly an order of magnitude in the absence of water vapor. It is therefore beneficial to add water to the ion flow tube to maintain an high instrument sensitivity at very dry conditions of the sampled air.

The water vapor dependence of the sensitivity can be parameterized by a third order polynomial (Eq. 2) for water vapor mixing ratios higher than 0.15 %. This is typically sufficient for atmospheric conditions. At lower water vapor mixing ratios as experienced in the chamber experiments the parameterization in Eq. 3 provides a good fit. S is the signal normalized by the primary ion, a, b, c, d are the fit parameters and $H_2O$ is the absolute water vapor mixing ratio.

$$S = a \times H_2O^3 + b \times H_2O^2 + c \times H_2O + d \qquad : \quad H_2O \geq 0.15\% \qquad (2)$$

$$S = c \times H_2O^{-0.4} + b \times H_2O + a \qquad : \quad H_2O < 0.15\% \qquad (3)$$

For the chamber experiments, the chamber air was humidified at the beginning of each experiments. At that time, no $HO_2$ is expected to be present in the chamber. Therefore, the increase in the background signal that has the same water vapor dependence as the sensitivity (see next section) can be used to determine the relative change of the sensitivity on water vapor on a daily basis. All $HO_2$ data from the chamber experiments shown in Sect. 3.4 were evaluated by applying this procedure.

During the series of chamber experiments presented in Sect. 3.4, calibrations were done in-between the experiments. In the middle of the series of experiments (6 June), settings of the instrument were tuned changing the sensitivity of the instrument. In total 6 calibrations were performed.

### 3.2 Precision of the $HO_2$ measurement

The precision of the instrument can be demonstrated by the Allan deviation plot shown in Fig. 4. 10 hours of measurement were used for this analysis while the instrument sampled from the calibration source that was operated at constant conditions. As mentioned above only the signal at mass-to-charge ratio 112 is used for simplicity. The calibration source constantly produced $2.5 \times 10^9$ $HO_2$ molecules cm$^{-3}$. A minimum integration time of 4 s was used for the evaluation, resulting in an Allan deviation of $1.7 \times 10^8$ $HO_2$ molecules cm$^{-3}$. With increasing integration time, the Allan deviation follows Gaussian noise demonstrating the statistically nature of the instrument's noise. An Allan deviation of $4.5 \times 10^7$ $HO_2$ molecules cm$^{-3}$ is achieved, if the measurement is averaged over 60 s. This is a sufficient detection limit for atmospheric measurements. Lower detection limits can be achieved, if, for example, an integration time of 10 min is acceptable. In addition, the use of both isotopic signals at mass-to-charge ratio 112 and 114 would lower the detection limit by a factor of $\sqrt{2}$.

### 3.3 Instrumental background

The instrumental background was characterized in experiments where the inlet was overflown with humidified synthetic air. This was done either using the radical source as a flow tube when the UV lamp was off or during experiments in SAPHIR, when only humidified synthetic air was present in the chamber. As shown in Fig. 5, the background signal changes similarly




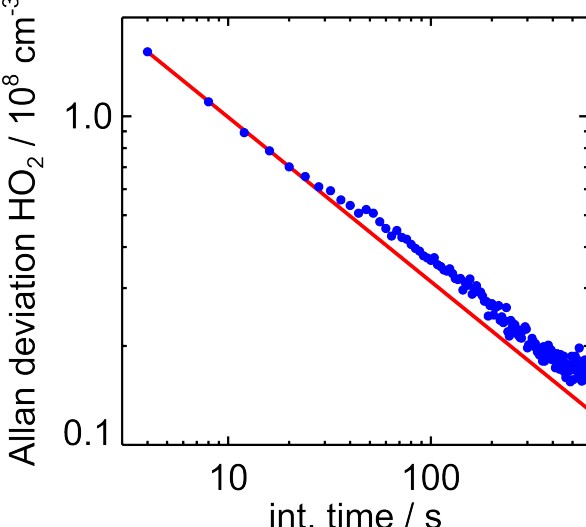

**Figure 4.** Allan deviation plot derived from sampling a constant $HO_2$ concentration of $2.5 \times 10^9$ $HO_2$ molecules $cm^{-3}$ over 10 hours. The Allan deviation demonstrates the precision of measurements depending on the integration time. The red line indicates the behavior of the Allan deviation, if the noise is only limited by Gaussian noise.

with water vapor for both experimental conditions. The shape of the water vapor dependence is consistent with the assumption that a constant $HO_2$ concentration ($1.5 \pm 0.2 \times 10^8$ molecules $cm^{-3}$) is internally produced in the instrument, which is detected according to the water vapor dependence of the instrument sensitivity discussed above. Therefore, the background can be be subtracted from the measured $HO_2$ concentration after applying the water vapor dependent calibration factor. The value of the background needs to be regularly determined. For chamber experiments reported here, the background signal was measured in the clean dark chamber at the start of each experiment.

In turn, the change in the background signal with changing water vapor reflects the relative change in the instrument sensitivity. This is especially relevant for the experiments in the SAPHIR chamber, because the chamber air was humidified starting from dry synthetic air at the start of the experiments. Once the water addition was started the signal was rising steep and decreased slightly at higher water concentration, as shown in Fig. 5. No trend of the background signal over a period of 2 month was observed. The day-to-day variability of the background (in total 16 experiments) was within a range of $\pm 12\,\%$ during 2 months of measurements at the chamber.

### 3.3.1 Potential interference from ozone

Ozone is known to be an interference in some $HO_2$ LIF instruments due to the photolysis of $O_3$ by the 308 nm excitation laser (Holland et al., 2003). The potential ozone interference in the CIMS $HO_2$ detection was investigated in laboratory experiments. Ozone was added to humidified synthetic air (water vapor mixing ratios 0.2 and 2.6 %). For both conditions no increase of the background signal could be observed for ozone mixing ratios of up to 400 ppbv.





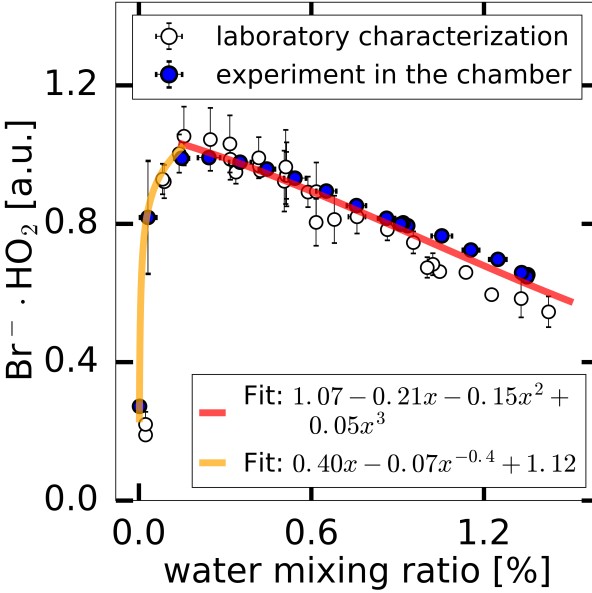

**Figure 5.** The background $HO_2$ measurement in the SAPHIR chamber derived during the humidification of the clean chamber and the background measured during the laboratory calibration supplying humidified synthetic air. The red line shows a third order polynomial fit representing Eg. 2, that can be used to corrected the instrument sensitivity at water mixing ratios higher than $0.15\,\%$. For lower mixing ratios the orange fit is used representing Eq. 3.

During experiments in the SAPHIR chamber, instrument background effects can only be determined for periods of the experiments without the presence of reactants, when no $HO_2$ was present. Typically, ozone was added in a concentration of $100$ to $200\,\mathrm{ppbv}$. Although no artefacts were found in the laboratory characterization, an increase in the background upon ozone addition was observed in two of 12 experiments in SAPHIR. For both experiments, the chamber was first humidified

5   and ozone was added afterwards. This appears as an increased intercept of $2.3 \times 10^8$ and $1.0 \times 10^8\,HO_2\,\mathrm{molecules\,cm^{-3}}$ in the linear regression between LIF and CIMS $HO_2$ data for the experiments of 21 June and 26 June (Fig. 7), respectively. The data of the LIF instrument were corrected for a maximum ozone interference of $0.05 \times 10^8$ and $0.15 \times 10^8\,HO_2\,\mathrm{molecules\,cm^{-3}}$ on these days, respectively. This correction is much smaller than the $HO_2$ concentration observed by the CIMS instrument, so that it can be excluded that differences are due to systematic errors in the data of the LIF instrument.

10   In the correlation plot (Fig. 8), including all experiments, this additional background was subtracted. The increased background due to the ozone addition has to be investigated in further chamber experiments. Because no direct connection between the occurrence of this interference and chemical conditions in the experiments is observed, it might be related to instrumental effects that could vary with time such as cleanness of the ion flow tube walls.





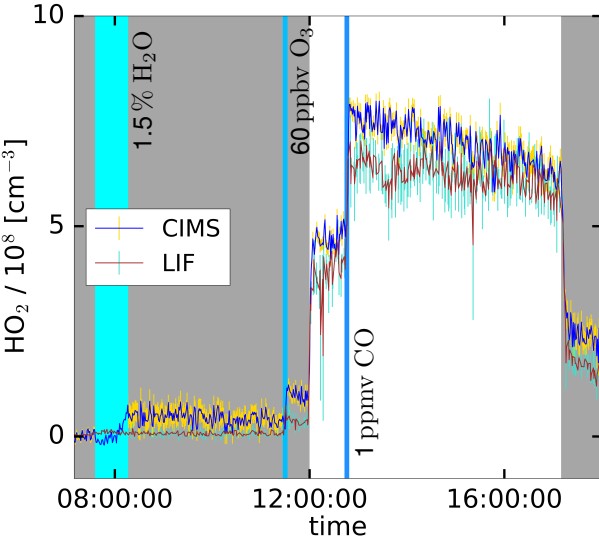

**Figure 6.** Time series plot for the $HO_2$ concentrations measured by the CIMS and the LIF instrument during the photo-oxidation experiment at 19 June 2017 in the SAPHIR chamber. The gray shaded area indicates that the chamber roof was closed. The vertical lines are showing the injection time of additional reactants, in case of water the injection took longer indicated by a broader line.

## 3.4 Comparison of CIMS and LIF $HO_2$ measurements

A time series for a typical experiment is shown in Fig. 6. The $HO_2$ production was initiated with the injection of ozone and the opening of the chamber roof providing UV light to the chamber. An addition of CO further boosted the $HO_2$ production, which dropped upon closing of the roof. However, $HO_2$ was still produced via radical chemistry in the dark. After the injection

of water the CIMS shows a stable signal with a small offset. During the experiment the LIF and CIMS data reveal a good correlation having similar errors. This experiment was performed without the addition of a volatile organic compound (VOC), as well as, two other experiments marked with "None" in Fig. 7.

Figure 7 displays the correlation between $HO_2$ measurements by the CIMS and the LIF instrument for all day-long photo-oxidation experiment in the SAPHIR chamber performed in this study. The chemical composition was varied between exper-

10 iments by changing for example the NO mixing ratio. The different chemical conditions during the experiments allows for checking for potential interferences. High NO concentrations of up to $3\,\mathrm{ppbv}$ were reached by injecting NO to the chamber air on 31.05 and 02.06, and up to $80\,\mathrm{ppbv}$ $NO_2$ was added on 23.06. The $NO_2$ interference test was performed injecting $NO_2$ in the dark, dry chamber. No further photo-chemistry experiments was done on this particular day. No systematic change in the relation between $HO_2$ data from the two instruments is observed in these cases (Fig. 7). In general, no interference from

15 VOCs (Isoprene, ISOPOOH and reaction products) are observed, except for experiments with IEPOX injections. IEPOX was detected on m/z 197 as $Br^- \cdot IEPOX$ ion cluster, but the instrument was not calibrated for IEPOX. Nevertheless, this mass trace can be used to correct the $HO_2$ measurement for the interference from IEPOX. The $HO_2$ signal observed during the



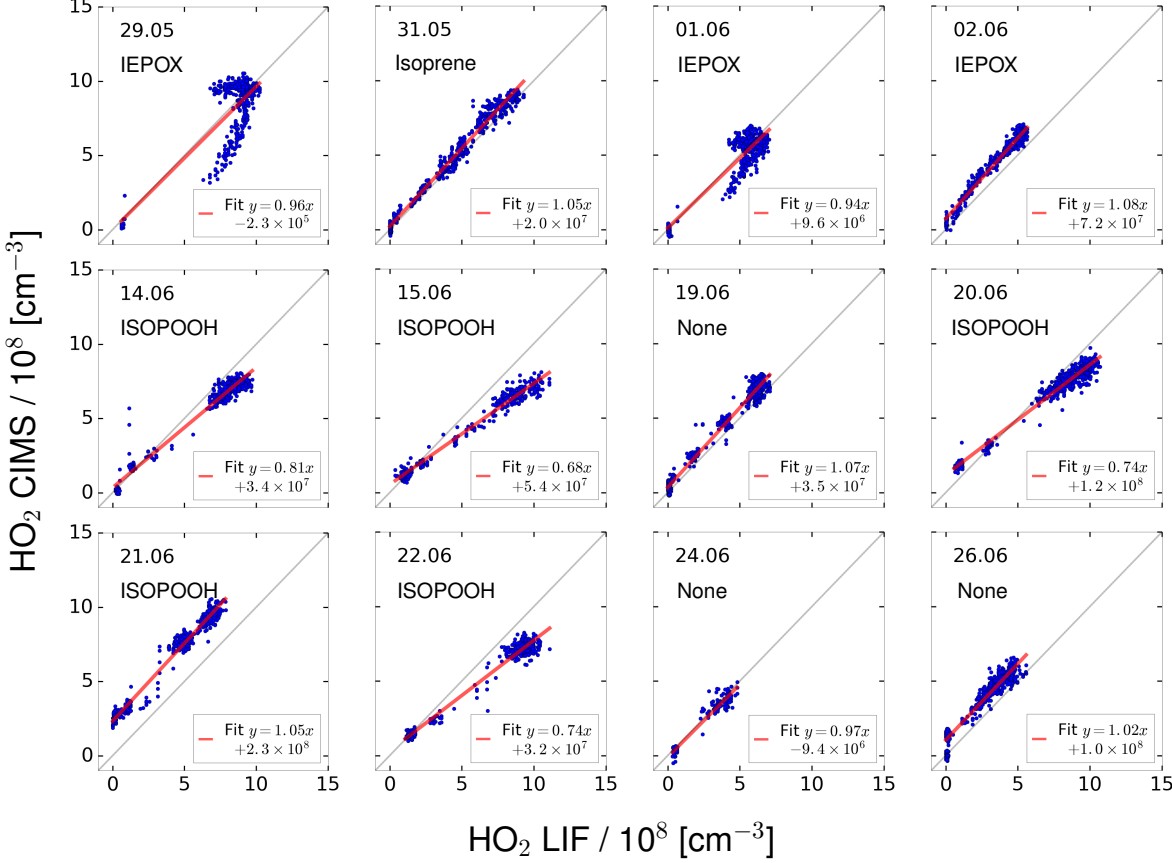

**Figure 7.** Correlation between $HO_2$ measurements by the CIMS and LIF instruments for individual chamber experiments. Labels in the plots indicate the specific VOC injected into the chamber.

injection of IEPOX can be attributed to the interference from IEPOX, because IEPOX was injected in the dark chamber so that no $HO_2$ is expected to be present. This gives the relationship between the signal observed at the IEPOX mass (m/z 197) to the interference signal from IEPOX at the $HO_2$ mass (m/z 112). During the photo-oxidation of IEPOX, when also $HO_2$ is present, the interference signal can be subtracted from the signal at the $HO_2$ mass by scaling the initial interference signal by the relative change on m/z 197. The correction improves the correlation of the CIMS and the LIF but the absolute agreement is still not as good (slope of the regression 0.86; coefficient of determination 0.79) compared to the other experiments. However, a correction was performed for all experiments with IEPOX injection. The corrections are in the order of or smaller than the $HO_2$ measurements, and works best for the experiment with the lowest IEPOX concentration. It is worth noting that IEPOX concentrations were at least 10 times higher than typically found in the atmosphere. Kaiser et al. (2016) found IEPOX concentrations of 1 ppbv during a campaign in a forest in the South-East US where isoprene, the precursor of IEPOX, was the





dominant organic species. Therefore no significant interference for atmospheric measurements by the CIMS instrument are expected from IEPOX.

During experiments with ISOPOOH, $HO_2$ measurements by the LIF instrument showed higher values than $HO_2$ measured by the CIMS instrument (slope of linear regression of 0.78; coefficient of correlation $R^2 = 0.68$). Further experiments will be

needed to investigate if ISOPOOH could cause an interference in the LIF instrument. Like in the case of IEPOX, ISOPOOH concentrations were much higher (several ppbv) than typically found in the atmosphere (less than 1 ppbv Kaiser et al. (2016)), so that no significant impact for atmospheric conditions is expected.

All concurrent measurements of the two instruments for $HO_2$ by CIMS and LIF, in the photo-oxidation experiments are summarized in the correlation plot shown in Fig. 8. In general, the correlation fit shows that there is an excellent agreement

of both instruments giving a slope of linear regression of 1.07 and the linear correlation coefficient $R^2$ is 0.87. Experiments investigating the photo-oxidation of IEPOX and ISOPOOH are color-coded and are excluded from the correlation fit. However, using all data for the correlation fit leads to similar result (slope of linear regression of 1.05; coefficient of correlation $R^2 =$ 0.89).

Correlation of individual experiments (Fig. 7, e.g. 21 June and 26 June) give partly significant offsets in the regression

analysis of up to $2.3 \times 10^8 \, \mathrm{cm}^{-3}$ $HO_2$. One possible reason could be the procedure, how the water vapor dependence of the instrument sensitivity was derived. This was done by using the relative change of the a presumably constant instrumental $HO_2$ background during the humidification of the clean chamber air. The water vapor concentration was measured at a different location in the chamber. Therefore, there is the potential that the water vapor concentration measurement in the chamber was not representative for the water concentration in the ion flow tube of the instrument. In this case, the determination of the

relative change of the instrument's sensitivity would fail and could results in an offset in the evaluation of data during the experiments. As seen in the experiment, shown in Fig. 6, there is a small offset that starts with the humidification. To avoid this effect in the future, a humidity sensor will be implemented at the ion flow tube.

## 4   Conclusion and Outlook

Chemical ionization was applied to measure atmospheric $HO_2$ concentrations using bromide ions as reagent. Laboratory

characterization experiments and measurements in the atmospheric simulation chamber SAPHIR in Jülich were used to check the instruments applicability for atmospheric measurements. The performance of the CIMS instrument is comparable with measurements by a laser-induced fluorescence instrument. A water vapor dependence of the instrument sensitivity needs to be taken into account in the evaluation of data because the sensitivity of the instrument changes by roughly a factor of 2 for atmospheric water vapor concentrations between 0.2 and 1.4 %. Also a water vapor dependent background signal is observed.

The change of the background signal with increasing water vapor, however, is explained by the water vapor dependence of the sensitivity. Therefore, the assumption is that the background consists of constant $HO_2$ production in the instrument. This background was stable within $\pm 12$ % during two months of measurements and no further trend was identified. The background signal and the instrument sensitivity needs to be quantified on a daily basis. No significant interference from trace gases





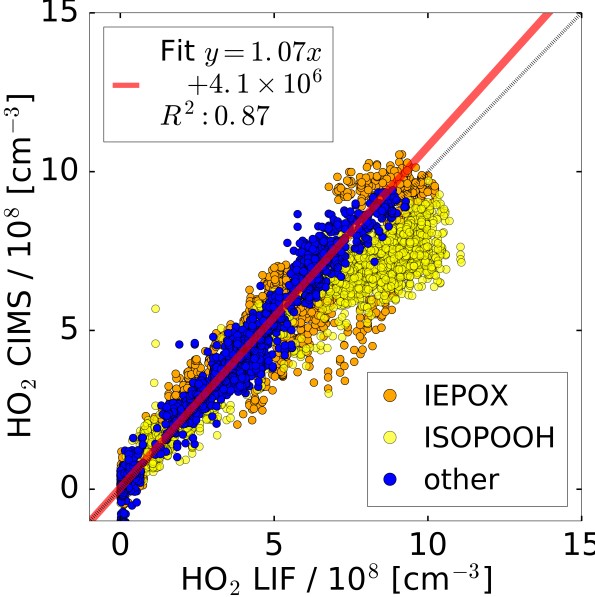

**Figure 8.** Correlation plot for the $HO_2$ concentrations measured by the CIMS and the LIF instrument of all photo-oxidation experiments in the SAPHIR chamber. A linear fit is applied to the subset of data excluding experiments with IEPOX and ISOPOOH.

NO, $NO_2$, $O_3$, CO, isoprene and ISOPOOH were found for atmospheric conditions. Only for non-atmospheric high IEPOX concentrations of several $\mathrm{ppbv}$ artificial signals were found that scaled with the IEPOX concentration. The $HO_2$ measurements correlate well with the LIF measurements. A slope of the linear regression of 1.07 was determined and a linear correlation coefficient ($R^2$) of 0.87 was found. With a detection limit of $4.5 \times 10^7 \,\mathrm{molecules\,cm^{-3}}$ for a $60\,\mathrm{s}$ measurement the instrument is suitable to measure typical $HO_2$ concentrations in the atmosphere.

Further improvements of the instrument sensitivity might be expected, if wall contact of the sampled air including $HO_2$ is further minimized. This could be achieved by applying a sheath flow of pure nitrogen along the surface of the ion flow-tube.

*Data availability.* Data of the experiments in the SAPHIR chamber used in this work is available on the EUROCHAMP data homepage (https://data.eurochamp.org/, last access: June 2018).





## Appendix A:  Experimental parameters

| day | $H_2O$ | $O_3$ | VOC | VOC conc. | NO | $NO_2$ | CO |
|---|---|---|---|---|---|---|---|
| **29 May** | 1.8 % | 140 ppbv | cis-IEPOX | 3.2 ppbv | 0.06 ppbv | 1.2 ppbv | 0.03 ppmv |
| **31 May** | 1.6 % | 50 ppbv | isoprene | 5 ppbv | 3.0 ppbv | 3.2 ppbv | 0.05 ppmv |
| **01 June** | 1.6 % | 140 ppbv | trans-IEPOX | 1.7 ppbv | 0.05 ppbv | 1.0 ppbv | 0.03 ppmv |
| **02 June** | 1.7 % | 30 ppbv | cis-IEPOX | 4.3 ppbv | 3.0 ppbv | 1.8 ppbv | 0.06 ppmv |
| **14 June** | 1.8 % | 170 ppbv | 1,2-ISOPOOH | 2.4 ppbv | 0.06 ppbv | 1.3 ppbv | 0.02 ppmv |
| **15 June** | 2.1 % | 190 ppbv | 1,2-ISOPOOH | 0.7 ppbv | 0.04 ppbv | 1.0 ppbv | 0.03 ppmv |
| **19 June** | 1.9 % | 70 ppbv | none | | 0.14 ppbv | 2.1 ppbv | 0.9 ppmv |
| **20 June** | 1.9 % | 160 ppbv | 1,2-ISOPOOH | 1.9 ppbv | 0.04 ppbv | 1.2 ppbv | 0.04 ppmv |
| **21 June** | 0.03 % | 170 ppbv | 1,2-ISOPOOH, | 6.5 ppbv | 0.01 ppbv | 0.4 ppbv | 0.15 ppmv |
| | | | 4,3-ISOPOOH | 1 ppbv | | | |
| **22 June** | 2.0 % | 170 ppbv | 4,3-ISOPOOH | 9 ppbv | 0.04 ppbv | 1.0 ppbv | 0.1 ppmv |
| **24 June** | 1.4 % | 160 ppbv | none | | 0.06 ppbv | 0.8 ppbv | 3.0 ppmv |
| **26 June** | 1.6 % | 170 ppbv | none | | 0.05 ppbv | 0.9 ppbv | 0.02 ppmv |

**Table A1.** The maximum concentration of reactants present in the reaction chamber during the experiments used for the correlation plots show in 7. The VOC concentrations for ISOPOOH and IEPOX are preliminary data and are having an higher uncertainty.

*Competing interests.*  The authors declare that they have no conflict of interest.

*Acknowledgements.*  This project has received funding from the European Research Council (ERC) (SARLEP grant agreement No. 681529) and from the European Research Council(EC) under the 15 European Union's Horizon 2020 research and innovation programme (Eurochamp 2020 grant agreement No.730997). Thanks to Patrick Veres from NOAA for discussion and first ideas regarding the $HO_2$ CIMS detection. We like to acknowledge Jean Rivera-Ros from the Harvard University and David Reimer from our institute for the synthesis of the VOCs used for this study. Thanks to Martin Breitenlechner, Alexander Zaytsev and Frank N. Keutsch from School of Engineering and Applied Sciences and Department of Chemistry and Chemical Biology, Harvard University, Cambridge, MA, USA for the PTR measurements performed.



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
