# Peer review of "Measurements of hydroperoxy radicals (HO2) at atmospheric concentrations using bromide chemical ionization mass spectrometry"

_Atmospheric Measurement Techniques, 2018_

## Referee Comment (RC1) · Anonymous Referee #1 · 3 Aug 2018

This paper presents a description of an instrument designed to detect HO2 radicals using bromide ion chemical ionization mass spectrometry. Most current methods of detecting HO2 radicals are indirect measurements utilizing chemical conversion of HO2 to OH by reaction with NO and subsequent detection of OH by laser-induced fluorescence in Fluorescence Assay by Gas Expansion (FAGE) instruments. Because this technique has been shown to suffer from interferences associated with the chemical conversion of RO2 radicals, a more direct method for detecting HO2 radicals would be a valuable addition to the community. While one could argue that the instrument described in this paper is not necessarily a "direct" method (page 3 line 6) as it still requires calibration, it is an important complimentary measurement technique. While

the work of Sanchez et al. (2016) provides some details of the technique, this paper provides more comprehensive measurements of potential interferences as well as an intercomparison with measurements from the more established LIF-FAGE technique.

The paper is well written and suitable for publication in AMT after the authors have addressed the following:

1) The authors describe the dependence of the sensitivity on water vapor concentration, with one possible explanation attributing the decrease in sensitivity to "The HO2 ion cluster is stabilized by water during the attachment process..." (page 7 line 6). Do the authors mean that during the Br- + HO2 attachment process, water vapor can stabilize the HO2- ion cluster, reducing the formation of the HO2 bromide ion cluster? This statement could use some clarification. (related "...access..." on line 7 should read "...excess...").

2) Similarly, the authors state that the formation of HO2-H2O clusters could also impact the sensitivity of the instrument, but the explanation is not clear. How does the formation of these clusters lead to a " roughly 10x increased in sensitivity at humid conditions" compared to dry conditions (page 8, line 6)? This should be clarified.

3) While the precision of the technique is described, the overall uncertainty in the CIMS measurement should be clarified, which I assume is primarily due to the uncertainty associated with the calibration technique.

4) The authors describe a background signal that appears to be a function of water vapor that may be due to production of HO2 inside the instrument, although possible mechanisms for production of the background are not discussed. The authors state that the measurements of the background are consistent with a constant value, and that the measured changes in the background signal with increasing water vapor shown in Figure 5 are consistent with a constant value for this background. The authors could provide additional support for this statement by converting the signals shown in Figure 5 to equivalent HO2 concentrations using the calibration factor's water dependence

shown in Figure 3.

5) Related to this, the paper would benefit from additional discussion of the nature of the background signal besides its dependence on water vapor. While it is reassuring that the measured background signal did not change over the course of these controlled experiments, understanding the nature of this background signal will be necessary to improve confidence in measurements in ambient air. Does the background vary with the strength of the ion source, pressure in the ion flow tube, inlet diameter, etc? Sanchez et al. (2016) also observed what appeared to be a constant background signal that they attributed to production of HO2 in their ion source. It appears that a similar signal is produced in this instrument, which should be discussed in more detail.

6) The authors also describe an ozone interference that appeared to occur in two of their experiments (page 11). Unfortunately, the source of this interference is not discussed in much detail, except to speculate that it may be related to instrumental effects or cleanliness of the ion tube walls. While the authors state that additional experiments will be needed to determine the source of this interference, the manuscript would benefit from an expanded discussion of this interference, including how it would have to be measured in ambient air.

7) For the correlation plots shown in Figure 7, the authors should state how the regression analysis was performed. They should perform bivariate regressions weighted by the precision of each measurement and should show the correlation coefficient on each plot. In particular, the correlations for the IEPOX experiments on 29.05 and 1.06 appear to be weak.

8) The authors also observe an interference with high concentrations of IEPOX, but there is little discussion of the cause of this interference and whether the authors expect similar interferences from other compounds under ambient conditions. Can the authors speculate on the mechanism of the interferences (decomposition of IEPOX inside the instrument)? It would be valuable to show the correlation with the LIF-FAGE

СЗ

measurements after the interference is subtracted from the IEPOX measurements in comparison with the measurements shown in Figure 7 (page 13 line 6).

9) While these chamber experiments illustrate the promise of the CIMS technique, additional measurements under ambient conditions will be necessary given the observation of several interferences in these experiments. This should be acknowledged in the manuscript. In particular, given the complex composition of ambient air, the authors should discuss strategies for testing for unknown interferences under ambient conditions, such as the addition of an ambient HO2 scavenger or other potential methods.

---

## Referee Comment (RC2) · Anonymous Referee #2 · 17 Aug 2018

This study investigated the application of a chemical ionization mass spectrometer using bromide ion as reagent for measurement of HO2 radicals. Calibration and characterization of the instrument were described. Experiments were conducted in the SAPHIR chamber using ISOPOOH, IEPOX, isoprene, and without injection of any VOC. The HO2 levels measured by the CIMS were compared to LIF and good agreements between the two instruments were observed.

In general, the manuscript is well-written and the topic will be of interest to the greater atmospheric community. The Br-CIMS technique for measuring HO2 was first developed in Sanchez et al. (2016), where the instrument capability was demonstrated with

measurements of ambient HO2 in an urban area. Here, the authors conducted an inter-comparison study with LIF-FAGE technique, which further validated the capability and the use of Br-CIMS in measuring HO2.

My major comments are that the authors should provide a clear focus of this work and put this work in the context of previously published results by Sanchez et al. A large portion of the manuscript (page 1-11) was on instrument characterization, covering ion flow tube specifications (pressure, residence time, etc), calibration procedure, sensitivity, water vapor dependence, detection limit, instrument background, and interference from ozone. All these are necessary and good as the way they were written, if there were no previously published work on the use of Br-CIMS for measuring HO2 radicals. However, all these were discussed in Sanchez et al. previously. But in this manuscript, there are no discussions at all regarding how these compare between this work and Sanchez et al. As this manuscript was submitted to an instrument journal, without such context, it is not clear if the aim of the study is to further improve the instrument beyond what was demonstrated previously (and if so, discuss the specific improvements), or if it is to directly adopt and reproduce what was in Sanchez et al., but with the main goal to compare the results to LIF. It is thus difficult for readers to evaluate whether the setup and performance of the instrument here is similar to or different from those in Sanchez et al. If the setup and performance are similar to Sanchez et al., this is a good thing, meaning that the measurement technique is robust and if others have an Aerodyne ToF-CIMS and want to measure HO2 they can also adopt this (fairly) readily. If the performance of instrument in this work is better than Sanchez et al., it is also a good thing, meaning that the technique has been further improved since then. However, it is difficult to tell from the current manuscript as no comparisons were made. The authors should compare the setup and relevant parameters to those in Sanchez et al. systematically and discussed accordingly. In addition to modifying the main text, a table documenting and comparing the various aspects of the instrument in the two studies would be very helpful.

It is interesting that the authors observed an instrument background, which was also reported in Sanchez et al. This should be discussed in more detail, as this was now observed in two independent studies and is intriguing/puzzling. In terms of the dependence of sensitivity on water vapor, the observed dependence is quite different between the two studies. The authors should mention this and discuss this accordingly.

Overall, the manuscript should be extensively modified to put this work in the context of previously published work, and to reflect the similarities/differences between the instrument setup and performance, etc. This would not only improve clarify of the manuscript but also help future researchers if they are interested in using or further developing this technique to measure HO2 radicals. I recommend the manuscript to be published after the authors address the major and specific comments.

Specific comments:

1. Page 3, line 6. Here in the introduction, the authors wrote "In this study, the direct measurement of atmospheric concentrations of HO2 radicals using Br-CIMS is presented. A detailed characterization of the instrument has been performed". This description also applies to what have been reported in Sanchez et al. I think this is a good place to set the tone for the manuscript and clearly describe the main focus/goal of the manuscript in the context of previously published work.

2. Page 3, line 22. It is noted that the mean residence time is 4ms. This seems very short (an order of magnitude too short?). Please show calculations on how the 4ms is obtained.

3. Page 3, line 31. What is the ion source (physically)?

4. Page 5, section 2.2. The manuscript flow will be improved if this section is deleted and the materials discussed here are added to section 3.1.

5. Page 6, line 25 onwards, calibration procedure.

a. In Figure 2, what does ncps stand for? I assume it is normalized cps. Please specify explicitly.

b. What is the cps of primary Br- (m/z 79)? Please specify explicitly.

c. If the m/z 112 signal (in cps) is normalized by the Br- signal (in cps), the y-axis in Figure 2 should be unit-less instead of ncps?

d. Considering all of the above, instead of using normalized ion count rates, I think it would be much easier for readers to interpret the data and compare the performance of Br-CIMS to prior work, if the authors can report the cps for m/z 112 and the cps for bromide ion. For instance, the range of [HO2] in this study (3e8 - 1.3e9 molecules/cc, i.e.,  $\sim$ 12-53 ppt) is similar to Sanchez et al., but it is hard to evaluate from Figure 2 if the sensitivity of the instrument is similar to or different from Sanchez et al.

e. Line 29. Is the unit for the slope correct?

6. Page 7-8, dependence of sensitivity on water mixing ratio.

a. Page 8, line 6. Please explain clearly how the "roughly 10x" higher sensitivity under humid conditions is determined.

b. Sanchez et al. observed a dependence of sensitivity for RH < 10%. The dependence of sensitivity on water mixing ratio in this study is quite different. More discussions are needed. Is there any difference in the instrument setup between this study and Sanchez et al. that could potentially lead to a different dependence?

7. Page 9-10, instrument background.

a. A constant background signal was also observed in Sanchez et al. Is the background observed here similar or different in magnitude compared to that in Sanchez et al.?

b. In Sanchez et al., the background signal does not scale with water mixing ratio (but scales linearly with UV lamp flux). Is there any difference in the setup that can lead to a different water mixing ratio dependence in the two studies? Does the background

СЗ

signal in this study scale with UV lamp flux? All these should be discussed.

c. It is suggested that there is a constant HO2 concentration produced in the instrument. What might be some potential sources? More discussions are needed.

d. Instead of A.U., it will help readers better interpret the data if the m/z 112 signals (in cps) are used in Figure 5. For instance, one cannot tell from Figure 5 if the magnitude of Br.HO2 is the same for laboratory characterization and experiments in the chamber.

e. It would be useful to show some mass spectra to help readers interpret the data. Please show mass spectra for 1) dry conditions for laboratory characterization, 2) humid conditions for laboratory characterization, 3) dry conditions for experiment in the chamber, 4) humid conditions for experiment in the chamber.

f. Was the temperature the same for laboratory characterization and experiment in the chamber?

8. Page 13 onwards. IEPOX interference.

a. Slopes and R2 values should be included in each subplot in Figure 7.

b. Line 1. Would be useful to show the relationship between the signal observed at the IEPOX mass (m/z 197) and the interference signal from IEPOX at the HO2 mass (m/z 112).

c. Line 5. Would be useful to show the correlations between CIMS and LIF with and without the correction.

d. Are the data in Figure 7 for IEPOX experiment corrected for the interference?

e. Page 14 line 14-22. The authors noted that the significant offsets for some experiments (e.g., 21 June and 26 June) could be due to how water vapor dependence of the instrument sensitivity was derived. However, the magnitude of the offset varied greatly from experiment to experiment. Why? Please discuss.

f. The "none" experiments in Figure 7 are experiments where no VOC was injected? However, the level of HO2 measured in these experiments was comparable to those experiments with 10s of ppb (?) of VOC? What is the source of HO2 radical in these "none" experiments? Shall the HO2 concentrations measured in the VOC-added experiments be corrected for this?

---

## Author Response (AR1)

Dear referees, Dear editor,

Thank you very much for your review of the paper.

**Response to referee #1:**

**Comment:**

1) The authors describe the dependence of the sensitivity on water vapor concentration, with one possible explanation attributing the decrease in sensitivity to "The $HO_2$ ion cluster is stabilized by water during the attachment process. . ." . Do the authors mean that during the $Br^-$ + $HO_2$ attachment process, water vapor can stabilize the $HO_2^-$ ion cluster, reducing the formation of the $HO_2$ bromide ion cluster? This statement could use some clarification. (related ". . .access. . ." on line 7 should read ". . .excess. . .").

**Response:**

The formation of an ion cluster needs an additional molecule that takes the excess energy upon the collision of the ion and the molecule forming the cluster. The formation of the hydrogen bond of the ion cluster would fail in most cases without leading away the collision energy. At atmospheric humidity virtual every ion comes with a loosely bound shell of water molecules. Therefore the following reaction seems favorable since water can take the collision energy.

$$Br^- \cdot H_2O + HO_2 \rightarrow Br^- \cdot HO_2 + H_2O \tag{R1}$$

In line 24 on page 8 the document was extended by a longer explanation that should provide clarity. "Two effects contribute to the water dependence: The initial increase of sensitivity (below 0.1% $H_2O$) comes from the stabilizing effect of $H_2O$. $Br^-$ adds $H_2O$, forming a loosely bound complex of $H_2O \cdot Br^-$; then, the $H_2O \cdot Br^-$ complex reacts with $HO_2$ according to the forward reaction R2. The steady decrease of sensitivity by a factor of 2 when the $H_2O$ mixing ratio is further increased to 1.2% comes from the back reaction of reaction R2."

**Comment:**

Similarly, the authors state that the formation of $HO_2^- \cdot H_2O$ clusters could also impact the sensitivity of the instrument, but the explanation is not clear. How does the formation of these clusters lead to a " roughly 10x increased in sensitivity at humid conditions" compared to dry conditions?

**Response:**

For completeness we discussed the formation of ion clusters and radical clusters with water that are involved in the process. The increase of the sensitivity is not related to the $HO_2 \cdot H_2O$ radical cluster, but the formation of the $HO_2 \cdot Br^-$ ion cluster is influenced by water. An $HO_2^- \cdot H_2O$ ion cluster was not part of the discussion. Because the additional discussion of radical cluster seems to cause some confusion we removed it.

**Comment:**

While the precision of the technique is described, the overall uncertainty in the CIMS measurement should be clarified, which I assume is primarily due to the uncertainty associated with the calibration technique.

**Response:**

Indeed, the uncertainty of the calibration makes the major contribution of the uncertainty of the measurement. This is $\pm 10\%$ ($1\sigma$) (Holland et al., 2003). A higher uncertainty might be introduced by the subtraction of the water vapor dependent background signal. The background signal was stable within $\pm 12\%$ ($1\sigma$) during the campaign.

Line 32 on page 10:

"Uncertainties are caused by the calibration, which makes the major contribution of the measurement uncertainty with $\pm 10\%$ ($1\sigma$) (Holland et al., 2003). The stability of the background signal in the measurements done here was $\pm 12\,\%$, giving an upper limit of the additional uncertainty from the stability of the subtracted background signal. Similar uncertainties are obtained by Sanchez et al. (2016)."

**Comment:**

The authors describe a background signal that appears to be a function of water vapor that may be due to production of $HO_2$ inside the instrument, although possible mechanisms for production of the background are not discussed. The authors state that the measurements of the background are consistent with a constant value, and that the measured changes in the background signal with increasing water vapor shown in Figure 5 are consistent with a constant value for this background.

The authors could provide additional support for this statement by converting the signals shown in Figure 5 to equivalent $HO_2$ concentrations using the calibration factor's water dependence shown in Figure 3.

Related to this, the paper would benefit from additional discussion of the nature of the background signal besides its dependence on water vapor. While it is reassuring that the measured background signal did not change over the course of these controlled experiments, understanding the nature of this background signal will be necessary to improve confidence in measurements in ambient air. Does the background vary with the strength of the ion source, pressure in the ion flow tube, inlet diameter, etc? Sanchez et al. (2016) also observed what appeared to be a constant background signal that they attributed to production of $HO_2$ in their ion source. It appears that a similar signal is produced in this instrument, which should be discussed in more detail.

**Response:**

Figure 5 has been changed accordingly.

Once the background is corrected for the water dependence sensitivity it seems to be constant as discussed in the paper. We assume that radicals are produced in the radioactive ion source. Unfortunately we cannot change the strength of the radioactive ion source. The pressure in the ion flow tube changes the overall sensitivity of the instrument and the background goes with that, so that it is not easy to differentiate between these effects. We will check for the nozzle diameter for future reference. For the chamber experiments and laboratory characterization experiments, when chemical conditions like presence of water vapor, ozone and $NO_x$ were much different or systematically varied, no dependence of the background on these parameters was observed.

Line 13 on page 11: "As reported for other CIMS instruments detecting radicals (Berresheim et al. (2000); Sanchez et al. (2016)), the radicals can be produced by the ion source. Therefore, this is the likely reason for the observed background signal. For chamber experiments reported here, the background signal was measured in the clean dark chamber at the start of each experiment."

**Comment:**

The authors also describe an ozone interference that appeared to occur in two of their experiments (page 11). Unfortunately, the source of this interference is not discussed in much detail, except to speculate that it may be related to instrumental effects or cleanliness of the ion tube walls. While the authors state that additional experiments will be needed to determine the source of this interference, the manuscript would benefit from an expanded discussion of this interference, including how it would have to be measured in ambient air.

**Response:**

The interference occurred upon ozone addition. However, this only happened in two of 16 experiments with ozone addition. The potential for a pure ozone interference was carefully tested in laboratory experiments, when the inlet was overflowed with ozone containing zero air and no ozone interference was found. The experiment is now shown in the Supplementary material. The cause of the increased background signals in the two experiments therefore is not clear and might not be directly related to ozone alone. At this point, we can only state that not all background signals can be clearly attributed to specific conditions, so that the background signal needs to be carefully characterized for the application of the instrument.

Line 12 on page 13: "This indicates that regular checks of the background signal is needed to take an appropriate background correction into account."

**Comment:**

For the correlation plots shown in Figure 7, the authors should state how the regression analysis was performed. They should perform bivariate regressions weighted by the precision of each measurement and should show the correlation coefficient on each plot. In particular, the correlations for the IEPOX experiments on 29.05 and 1.06 appear to be weak.

**Response:**

We changed the plot and added to the caption of Fig. 7: "For the regression line shown in blue a least square fit has been performed, the errors $(1\sigma)$ of the measurement are indicated in gray."

Further we added on page 14 in line 10: "The results of a linear regression analysis are given in Fig. 7, which takes errors in both $HO_2$ measurements into account (Press et al., 1992)."

**Comment:**

The authors also observe an interference with high concentrations of IEPOX, but there is little discussion of the cause of this interference and whether the authors expect similar interferences from other compounds under ambient conditions. Can the authors speculate on the mechanism of the interferences (decomposition of IEPOX inside the instrument)?

**Response:**

Since no other significant interferences by VOCs was observed, the only plausible explanation seems to be the fragmentation of the molecule in the transfer stage. The fragmentation there can be initiated by acceleration of the ions in the electrostatic field causing collisions with other molecules.

Line 17 on page 15: "A plausible reason for the IEPOX interference found seems to be a fragmentation of the cluster ion in the transfer stage of the instrument. The fragmentation could be initiated by acceleration of the ions in the electrostatic field causing collisions with other molecules."

**Comment:**

It would be valuable to show the correlation with the LIF-FAGE measurements after the interference is subtracted from the IEPOX measurements in comparison with the measurements shown in Figure 7.

**Response:**

The figure already shows data corrected for IEPOX.

Line 9 on page 15: "The correlation plots shown in Fig. 7 are corrected for the IEPOX interference."

**Comment:**

While these chamber experiments illustrate the promise of the CIMS technique, additional measurements under ambient conditions will be necessary given the observation of several interferences in these experiments. This should be acknowledged in the manuscript. In particular, given the complex composition of ambient air, the authors should discuss strategies for testing for unknown interferences under ambient conditions, such as the addition of an ambient $HO_2$ scavenger or other potential methods.

**Response:**

The chamber experiments were the first tests of the applicability of the instrument for ambient $HO_2$ concentration measurements. It allows comparing the measurements to the FAGE-LIF measurements under controlled and defined conditions and ensures that both instruments sample the exactly the same air mass. Although the chemical composition of ambient is more complex, the most important known constituents of the atmosphere are present in the photochemical experiments in the SAPHIR chamber. Such a comparison would be more uncertain in a field experiment due to the inhomogeneities of the air mixture. Potential interferences of most abundant species like $NO_x$, $O_3$, ambient concentrations of VOCs were tested in the chamber experiments. Deployment of the instrument in the field is the next step.

Scavenging of $HO_2$ might not be as easy it looks at first glance. The addition of high NO concentrations would convert $HO_2$ to OH, but could also lead to secondary chemistry in the ion flow tube that could cause other artefacts. Such a scheme is worth trying, but would require extensive characterization and testing. The strategy of comparing measurements of two independent instruments applying different methods is another strategy to identify interference assuming that interferences would not have the identical effect on both instruments. It is foreseen for the future application that $HO_2$ is concurrently detected by the LIF and CIMS instruments.

Added line 14 on page 17: "Chemical conditions in the chamber experiments were close to atmospheric conditions regarding the most important constituents of the atmosphere such as $NO_x$, ozone and water vapor showing the applicability of the instrument under these conditions. First future deployment in field experiments will be done with concurrent $HO_2$ measurements by the LIF instrument, so that potential so far unrecognized interference can be identified. "

**Response to referee #2:**

**Major comment:**

My major comments are that the authors should provide a clear focus of this work and put this work in the context of previously published results by Sanchez et al. (2016). A large portion of the manuscript (page 1-11) was on instrument charac-
terization, covering ion flow tube specifications (pressure, residence time, etc), calibration procedure, sensitivity, water vapor dependence, detection limit, instrument background, and interference from ozone. All these are necessary and good as the way they were written, if there were no previously published work on the use of Br-CIMS for measuring $HO_2$ radicals. However, all these were discussed in Sanchez et al. (2016) previously. But in this manuscript, there are no discussions at all regarding how these compare between this work and Sanchez et al. (2016). As this manuscript was submitted to an instrument journal,
without such context, it is not clear if the aim of the study is to further improve the instrument beyond what was demonstrated previously (and if so, discuss the specific improvements), or if it is to directly adopt and reproduce what was in Sanchez et al. (2016), but with the main goal to compare the results to LIF. It is thus difficult for readers to evaluate whether the setup and performance of the instrument here is similar to or different from those in Sanchez et al. (2016). If the setup and performance are similar to Sanchez et al. (2016), this is a good thing, meaning that the measurement technique is robust and if others have
an Aerodyne ToF-CIMS and want to measure $HO_2$ they can also adopt this (fairly) readily. If the performance of instrument in this work is better than Sanchez et al. (2016), it is also a good thing, meaning that the technique has been further improved since then. However, it is difficult to tell from the current manuscript as no comparisons were made. The authors should compare the setup and relevant parameters to those in Sanchez et al. systematically and discussed accordingly. In addition to modify-
ing the main text, a table documenting and comparing the various aspects of the instrument in the two studies would be very helpful. It is interesting that the authors observed an instrument background, which was also reported in Sanchez et al. (2016). This should be discussed in more detail, as this was now observed in two independent studies and is intriguing/puzzling. In terms of the dependence of sensitivity on water vapor, the observed dependence is quite different between the two studies. The authors should mention this and discuss this accordingly. Overall, the manuscript should be extensively modified to put this work in the context of previously published work, and to reflect the similarities/differences between the instrument setup and
performance, etc. This would not only improve clarify of the manuscript but also help future researchers if they are interested in using or further developing this technique to measure $HO_2$ radicals. I recommend the manuscript to be published after the authors address the major and specific comments.

**Response:**

Indeed Sanchez et al. (2016) provided pioneer work with the presentation of the bromide CIMS. The aim of our paper
is further characterization of the technique and the search for possible interferences. Especially the comparison with the $HO_2$ LIF allows quantitatively comparing $HO_2$ measurements with a reference and the identification of potential interferences under realistic atmospheric conditions. Further we successfully improved the instrument's sensitivity for $HO_2$ employing our custom-build ion flow tube. In the revised version of the manuscript we include comparisons with the instrument described in Sanchez et al. (2016) in more detail (see responses to the further comments).

**Comment:**

It is interesting that the authors observed an instrument background, which was also reported in Sanchez et al. (2016). This should be discussed in more detail, as this was now observed in two independent studies and is intriguing/puzzling. In terms of the dependence of sensitivity on water vapor, the observed dependence is quite different between the two studies. The authors should mention this and discuss this accordingly.

**Response:**

We added line 3 on page 12: "Sanchez et al. (2016) also described a constant $HO_2$ source which causes a background. An $HO_2$ titration experiment Sanchez et al. (2016) confirmed that $HO_2$ is internally produced, which has been discussed for other radical measurements using a CIMS approach (Berresheim et al., 2000)."

Sanchez et al. (2016) found a constant sensitivity for water vapor mixing ratios between 0.2 and 0.8% whereas a 30%
decrease is observed here. Only for one sensitivity measurement at 0.06% water mixing ratios an increased sensitivity by approximately 50% is reported by Sanchez et al. (2016). The reason for this different behaviour is not clear, but one may speculate that the design of the ion-flow tube and inlet nozzle might impact the collision probability of ion clusters. The relative change of the instrument's sensitivity in Sanchez et al. (2016) towards dry conditions is not reported, so that it is not clear, if the sensitivity drops for dry conditions in their instruments as observed here.

**Comment:**

Page 3, line 6. Here in the introduction, the authors wrote "In this study, the direct measurement of atmospheric concentrations of $HO_2$ radicals using Br-CIMS is presented. A detailed characterization of the instrument has been performed". This description also applies to what have been reported in Sanchez et al. (2016). I think this is a good place to set the tone for the manuscript and clearly describe the main focus/goal of the manuscript in the context of previously published work.

**Response:**

We added in line 8 on page 3: "Sanchez et al. (2016) demonstrated that the most promising ionization technique is the detection of the bromide cluster with $HO_2$. In their work they showed that a sufficient sensitivity can be achieved and no significant interference from $NO_x$, HCHO, $SO_2$, $O_3$ is present. Based on the work of Sanchez et al. (2016) a custom-built ionization flow tube optimized for the sampling of radicals was mounted on top of an Aerodyne TOF mass spectrometer for the detection of $Br \cdot HO_2$ clusters in this work. In addition to laboratory characterization experiments that mostly confirmed results reported in Sanchez et al. (2016), the performance of the instrument was quantitatively assessed in a comparison of $HO_2$ concentrations with measurements by an established $HO_2$ instrument using laser-induced fluorescence. Experiments in the atmospheric simulation chamber SAPHIR were performed at atmospheric gas mixtures and radical concentrations. "

**Comment:**

Page 3, line 30. It is noted that the mean residence time is 4ms. This seems very short (an order of magnitude too short?). Please show calculations on how the 4ms is obtained.

**Response:**

Indeed this was an error, the conversion of minutes to seconds was missing. The text has been corrected for the correct time, which is 240 ms.

**Comment:**

What is the ion source (physically)?

**Response:**

It is a foil coated with radioactive $^{210}$Po having an activity of 370 MBq (equal to 10 mCi) contained in a sealed tube. The type and manufacturer is mentioned in the text (Page 4, line 7) now.

**Comment:**

Page 5, section 2.2. The manuscript flow will be improved if this section is deleted and the materials discussed here are added to section 3.1.

**Response:**

Section 2.2 has been moved further down followed by section 3.1.

**Comment:**

In Figure 2, what does ncps stand for? I assume it is normalized cps. Please specify explicitly.

**Response:**

Yes, in the figures caption it is mentioned that the normalized count rate is shown. However, for clarity we removed ncps.

**Comment:**

What is the cps of primary Br- (m/z 79)? Please specify explicitly.

**Response:**

We added the information to section 3.2.

**Comment:**

If the m/z 112 signal (in cps) is normalized by the Br- signal (in cps), the y-axis in Figure 2 should be unit-less instead of ncps?

**Response:**

Basically this is the definition of the normalized signal, it is unit-less, for clarity we removed ncps.

**Comment:**

Considering all of the above, instead of using normalized ion count rates, I think it would be much easier for readers to interpret the data and compare the performance of Br-CIMS to prior work, if the authors can report the cps for m/z 112 and the cps for bromide ion. For instance, the range of [$HO_2$] in this study ($3\times10^8 - 1.3\times10^9$ molecules/cc, i.e., 12-53 ppt) is similar to Sanchez et al. (2016), but it is hard to evaluate from Figure 2 if the sensitivity of the instrument is similar to or different from Sanchez et al. (2016).

**Response:**

This is now discussed for comparison of both papers (Page 17 Line 9). " $HO_2$ was directly sampled through a nozzle into a custom-build ion flow tube which was optimized for sensitivity. The sensitivity reached is equal to $0.005 \times 10^8$ $HO_2$ per $cm^3$ for $10^6$ cps of bromide and 60 s of integration time, which is approximately 3 times higher than the sensitivity for a similar instrument by Sanchez et al. (2016)."

**Comment:**

Page 7 Line 9. Is the unit for the slope correct?

**Response:**

Indeed it should be [$cm^3$].

**Comment:**

Please explain clearly how the "roughly 10x" higher sensitivity under humid conditions is determined.

**Response:**

The ozonolysis experiment explained in this section provides the possibility to compare measurements with and without humidity. Comparing the sensitivity determined without water addition and the sensitivity at 0.1% water mixing ratio gives a roughly a factor of 10. As a response of the comment of reviewer 1, this sentence was removed from this position. We modified the sentence on page 8 in line 21: "For water vapor mixing ratio of less than 0.1 %, the sensitivity drops quickly by a factor of 7 at dry conditions compared to the maximum sensitivity at 0.1 % water vapor mixing ratio."

**Comment:**

Sanchez et al. (2016) observed a dependence of sensitivity for RH < 10%. The dependence of sensitivity on water mixing ratio in this study is quite different. More discussions are needed. Is there any difference in the instrument setup between this study and Sanchez et al. (2016) that could potentially lead to a different dependence?

**Response:**

Sanchez et al. (2016) found a constant sensitivity for water vapor mixing ratios between 0.2 and 0.8% whereas a 30% decrease is observed here. Only for one sensitivity measurement at 0.06% water mixing ratios an increased sensitivity by approximately 50% is reported by Sanchez et al. (2016). The reason for this different behaviour is not clear, but one may speculate that the design of the ion-flow tube and inlet nozzle might impact the collision probability of ion clusters. The relative change of the instrument's sensitivity in Sanchez et al. (2016) towards dry conditions is not reported, so that it is not clear, if the sensitivity drops for dry conditions in their instruments as observed here.

**Comment:**

A constant background signal was also observed in Sanchez et al. (2016). Is the background observed here similar or different in magnitude compared to that in Sanchez et al. (2016)?

**Response:**

Sanchez et al. (2016) determined an instrument background of 4 pptv or more, since the scavenging is not efficient in the ion flow tube. This seems to be in a similar magnitude. We found a background of 6 pptv during the chamber experiments.

We added at page 12 in line 5: "Sanchez et al. (2016) determined an instrument background of at least 4 pptv $HO_2$, which compares well with the background of 6 pptv $HO_2$ that has been found during the experiments in the SAPHIR chamber."

**Comment:**

In Sanchez et al. (2016), the background signal does not scale with water mixing ratio (but scales linearly with UV lamp flux). Is there any difference in the setup that can lead to a different water mixing ratio dependence in the two studies? Does the background signal in this study scale with UV lamp flux? All these should be discussed.

**Response:**

The effect of an increased background signal during the calibration procedure in Sanchez et al. (2016) is clearly connected to their calibration source and not to the instrument's performance, because it scales with the intensity of the UV lamp that is part of the calibration source. We would expect that this behavior does not apply for ambient air measurements of the instrument in Sanchez et al. (2016). Such an effect of the calibration source has not been observed for our calibration source that is the same as has been used for the calibration of the $HO_2$ LIF instrument for more than 20 years without any hint that there is $HO_2$ or interference signal produced in the absence of water. One difference in the calibration procedure is that Sanchez et al. (2016) worked at very low humidity to produce ambient $HO_2$ concentrations, whereas our calibration source is operated at ambient humidity. In addition they work with purified air, whereas our calibration source is operated with clean synthetic air (purity

99.9999%). One could only speculate that photolysis of impurities at 185 nm might be the cause for the higher background observed by Sanchez et al. (2016), but we do not think that this needs to be discussed in this paper.

**Comment:**

It is suggested that there is a constant $HO_2$ concentration produced in the instrument. What might be some potential sources? More discussions are needed.

**Response:**

The only potential source of radicals in the instrument is the ion source. Please compare the answer to a similar comment of reviewer#1, where more discussion about this topic is cited.

**Comment:**

Instead of A.U., it will help readers better interpret the data if the m/z 112 signals (in cps) are used in Figure 5. For instance, one cannot tell from Figure 5 if the magnitude of $Br \cdot HO_2$ is the same for laboratory characterization and experiments in the chamber.

**Response:**

As suggested by referee#1, the Figure shows the background in $HO_2$ equivalents now. During the laboratory experiments we found an up to 20% lower background.

**Comment:**

It would be useful to show some mass spectra to help readers interpret the data. Please show mass spectra for 1) dry conditions for laboratory characterization, 2) humid conditions for laboratory characterization, 3) dry conditions for experiment in the chamber, 4) humid conditions for experiment in the chamber.

**Response:**

There are no significant difference between mass spectra of the two masses at which $HO_2 \cdot$ Br cluster appear (112 / 114) for the conditions that are mentioned by the reviewer. Humidity only affects the sensitivity (=count number), but not the shape of the spectrum. We add a mass spectrum in the supplementary material.

**Comment:**

Was the temperature the same for laboratory characterization and experiment in the chamber?

**Response:**

This is a good point, we added on page 6 in line 12: "Measurements in the chamber were performed at daytime temperatures of roughly 20 to 30 ° Celsius. Additionally, the instrument itself was temperature stabilized to 25±5 ° Celsius to prevent temperature effects.

And on page 4 in line 2: Laboratory experiments were performed at 25 to 30 ° Celsius."

**Comment:**

Slopes and R2 values should be included in each subplot in Figure 7.

**Response:**

The slope is already included in Figure 7, R2 will be added.

**Comment:**

Would be useful to show the relationship between the signal observed at the IEPOX mass (m/z 197) and the interference signal from IEPOX at the $HO_2$ mass (m/z 112).

**Response:**

Figures have been added to the supplementary material for this purpose.

**Comment:**

Would be useful to show the correlations between CIMS and LIF with and without the correction.

**Response:**

The contribution of the interference from the IEPOX is highly variable in the experiments. When the IEPOX is introduced into the chamber, no $HO_2$ was present, but IEPOX concentrations decreased to nearly zero within a few hours. Therefore, we think that the new figures in the supplement showing the $HO_2$ equivalent signal of the interference as time series is most useful for the reader. A correlation plot of the uncorrected signal does not contain any additional information.

**Comment:**

Are the data in Figure 7 for IEPOX experiment corrected for the interference?

**Response:**

Yes, this is now explained in the text.

**Comment:**

The authors noted that the significant offsets for some experiments (e.g., 21 June and 26 June) could be due to how water vapor dependence of the instrument sensitivity was derived. However, the magnitude of the offset varied greatly from experiment to experiment. Why? Please discuss.

**Response:**

The offset of the linear regression is not very accurate, if the dynamic range of the data set is small compared to the scatter of data. This explains partly the variability of the offset in the regression of individual experiments. For the determination of the offset, the signal derived during the humidification process was used. This means that water vapor from boiling water is introduced with a high flow of synthetic air, so that humidity changes quickly and air masses might not be perfectly mixed. Only for these conditions, there is potential that the water vapor measurement in the chamber could differ from the water concentration sampled by the CIMS instrument. This not ideal determination of the water vapor dependent background will be avoided in future application.

We added changes on page 16 Line 1: "This was done by using the measured signal at the $HO_2 \cdot Br^-$ mass during the humidification process of the clean chamber air, when no $HO_2$ was present. However, the chamber air might not be perfectly mixed during the humidification, because water vapor from boiling water is introduced at one location in the chamber together with a high flow of synthetic air. Because the water measurement in the chamber used for the determination of the CIMS background signal and the CIMS inlet are at different locations in the chamber, the water measurement is potentially not accurate for the water vapor sampled by the CIMS for these conditions, so that small systematic errors in the background determination cannot be excluded. In the future, the water vapor dependence of the background will be determined independently from the chamber experiment, so that it can be expected that such effects will not be relevant."

**Comment:**

The "none" experiments in Figure 7 are experiments where no VOC was injected? However, the level of $HO_2$ measured in these experiments was comparable to those experiments with 10s of ppb of VOC? What is the source of $HO_2$ radical in these "none" experiments? Shall the $HO_2$ concentrations measured in the VOC-added experiments be corrected for this?

**Response:**

Indeed, the label indicates which VOC was added. The experiments are all photo-chemistry in synthetic air. In the photochemistry experiments with no addition of OH reactants, $HO_2$ is present, because (1) OH and NO are produced from the photolysis of HONO that is released from the Teflon film of the chamber and (2) small concentrations of OH reactants are present converting OH to $RO_2$ and $HO_2$. Details of experiments without the addition of OH reactants can be found in Rohrer et al. (2005). The $HO_2$ measured under these conditions is $HO_2$ and not an interference. Often $HO_2$ is well explained assuming that the measured OH reactivity produces directly $HO_2$ (see for example Fuchs et al. (2013)). The good agreement of $HO_2$ measurements by both instruments, LIF and CIMS, confirms that the $HO_2$ is indeed present in the experiment.

For clarity we added on page 14 Line 5: "Nevertheless, $HO_2$ is produced in these experiments, because OH and NO are produced from the photolysis of HONO released from the chamber Teflon film in the sunlit chamber (Rohrer et al., 2005). Reaction of small concentrations of OH reactants formed under these conditions in the chamber lead to the formation of $HO_2$ (Rohrer et al., 2005)."

[revised manuscript text omitted]

5   can lead to the concurrent conversion of $RO_2$.

Previous work by Veres et al. (2015) showed that $HO_2$ radicals can be detected with a CIMS instrument using iodide as primary ion. Sanchez et al. (2016) demonstrated for the first time that this approach can also be used with $Br^-$. $HO_2$ radicals are directly measured by a mass spectrometer as an ion cluster formed with bromide ions. Sanchez et al. (2016) demonstrated that the most promising ionization technique is the detection of the bromide cluster with $HO_2$. In their work they showed

10   that a sufficient sensitivity for atmospheric measurements can be achieved and no significant interference from $NO_x$, $HCHO$, $SO_2$, $O_3$ is present. Following the concept of Sanchez et al. (2016) a bromide chemical ionization mass spectrometer with improved sensitivity was developed in this work. An optimized ionization flow tube was custom built and mounted on top of a commercial, high resolution time-of-flight mass spectrometer (TOF-MS, Aerodyne Res.). In addition to laboratory charac- terization experiments that mostly confirmed results reported in Sanchez et al. (2016), the performance of the instrument was

15   quantitatively assessed in a comparison of $HO_2$ concentrations with measurements by an established $HO_2$ instrument using laser-induced fluorescence. Experiments in the atmospheric simulation chamber SAPHIR were performed at atmospheric gas mixtures and radical concentrations.

**2   Methods**

**2.1   Chemical ionization mass spectrometry technique**

[revised manuscript text omitted]

5 Sect. 2.4. Figure 2 shows the measured, normalized ion count rates measured by the CIMS, when the calibration source was operated at a constant water vapor mixing ratio of $1.0\,\%$. The $HO_2$ concentration was varied by changing the UV radiation intensity by varying the $N_2O$ concentration in the absorption cell of the calibration source. A linear behavior for the normalized count rate measured by the CIMS instrument is observed in the tested range of $3.0 \times 10^8$ to $1.3 \times 10^9$ $HO_2$ molecules cm$^{-3}$. The slope of the linear regression gives the calibration factor of $6.8 \times 10^{-12}$ cm$^3$. The intercept of $5.1 \times 10^{-4}$ of the linear fit

10 indicates a $HO_2$ background signal that was not corrected in Fig. 2.

**3.2 Instrument sensitivity**

The possible dependence of the $HO_2$ detection sensitivity on the concentration of gaseous water vapor mixing ratio was studied using two different radical sources. The water dependent calibration factor is defined by Eq. 1, where $c$ represents the instrument sensitivity that depends on the water concentration.

$$\quad \frac{m/z(112)}{m/z(79)} = c(H_2O) * [HO_2] \tag{1}$$

One of the radical sources is described in Sect. 2.4. Keeping the UV flux of the photolysis lamp constant, different $HO_2$ concentrations were produced by varying the water-vapor mixing-ratio between 0.1 - 1.2%. As the $HO_2$ concentration provided by the calibration can be accurately calculated for different water mixing ratios, the influence of water on the $HO_2$ detection sensitivity could be investigated. Measurements at dry conditions were not possible, because the calibration source needs water 10 to generate $HO_2$.

For low water vapor concentrations, ozonolysis of 2,3 dimethyl-2-butene was used as a radical source. For that purpose, the alkene was added in a concentration of 30 ppbv to a mix of synthetic air and 200 ppbv ozone. The radical source (with photolysis lamp switched off) was used as a flow-tube to overflow the inlet of the instrument with this gas mixture. 0.2% CO was added to scavenge OH radicals produced from the ozonolysis reaction by a fast conversion of OH to $HO_2$. The water 15 mixing ratio was altered during the ozonolysis experiment from 0.0 to 0.6%. Assuming that the $HO_2$ concentration from the ozonolysis is constant, the relative change in the signal gives the relative change of the instrument sensitivity. Absolute sensitivities were derived by scaling the $HO_2$ signals from the ozonolysis experiment to the concentration derived by the water dependent calibration from the radical source by multiplication with a constant factor.

Figure 3 shows the sensitivity determined for each water vapor mixing ratio showing a decreasing sensitivity with increasing 20 water vapor mixing ratio for atmospheric relevant water mixing ratios higher than 0.1 %. The water dependent decrease in sensitivity is nearly linear for this range of water vapor mixing ratios. For water vapor mixing ratio of less than 0.1 %, the sensitivity drops quickly by a factor of 7 at dry conditions compared to the maximum sensitivity at 0.1 % water vapor mixing ratio.

Two effects contribute to the water dependence: The initial increase of sensitivity (below 0.1% $H_2O$) comes from the sta-25 bilizing effect of $H_2O$. $Br^-$ adds $H_2O$, forming a loosely bound complex of $H_2O \cdot Br^-$; then, the $H_2O \cdot Br^-$ complex reacts with $HO_2$ according to the forward reaction R2. The steady decrease of sensitivity by a factor of 2 when the $H_2O$ mixing ratio is further increased to 1.2% comes from the back reaction of reaction R2.

$$HO_2 + Br^- \cdot H_2O \rightleftharpoons Br^- \cdot HO_2 + H_2O \tag{R2}$$

The water vapor dependence of the sensitivity can be parameterized by a third order polynomial (Eq. 2) for water vapor 30 mixing ratios higher than 0.1 %. This is typically sufficient for atmospheric conditions. At lower water vapor mixing ratios

[Figure]

**Figure 3.** Measured $HO_2$ sensitivity as a function of the water mixing ratio in two experiments. For the calibration, $HO_2$ was produced by the radical source while varying the water vapor concentration which causes a change in the $HO_2$ radical concentration. During the ozonolysis experiment, $HO_2$ was produced from the ozonolysis of 2,3-dimethyl-2-butene, which is independent of the water vapor mixing ratio. The red line shows a third order polynomial fit applied to the calibration data for the range of water vapor mixing ratios higher than 0.1 %.

the parametrisation in Eq. 3 provides a good approximation. Such low water vapor mixing ratios were present in the chamber experiments after flushing the chamber before an experiment started.

$$S = a \times H_2O^3 + b \times H_2O^2 + c \times H_2O + d \qquad : \quad H_2O \geq 0.1\% \tag{2}$$

$$S = c \times H_2O^{-0.4} + b \times H_2O + a \qquad : \quad H_2O < 0.1\% \tag{3}$$

5   S is the signal normalized by the primary ion, a, b, c, d are the fit parameters and $H_2O$ is the absolute water vapor mixing ratio. During the series of chamber experiments presented in Sect. 3.5, calibrations were done in-between the experiments. In the middle of the series of experiments (06 June), settings of the instrument were tuned changing the sensitivity of the instrument. In total 6 calibrations were performed.

To gain sensitivity the wall contact was reduced by directly sampling via a nozzle into the ion flow tube in the instrument
10  here. The ion flow tube was further optimized for length and pressure to improve the sensitivity for $HO_2$. Basically the ion flow tube used during this study (130 mm length) was compared to a similar ion flow tube with a length of 200 mm. However, this resulted in 50 % less sensitivity at 120 hPa, which has been identified as the optimal pressure in terms of sensitivity. Finally

the flows were optimized to gain the maximum in sensitivity. Further sensitivity can be gained combining both isotopic signals for the data analysis as already mentioned by Sanchez et al. (2016).

For the chamber experiments, the chamber air was humidified at the beginning of each experiment. At that time, no $HO_2$ is expected to be present in the chamber. Therefore, the signal caused by the constant $HO_2$ background changes with the water vapor dependence of the instrument sensitivity (see next section) and could be used to determine the relative change of the sensitivity on water vapor for an individual experiment during this measurement campaign. All $HO_2$ data from the chamber experiments shown in Sect. 3.5 were evaluated by applying this procedure.

As shown in Fig. 3, the instrument response to the change of the water vapor concentration is similar both methods of radical production. In addition, the instrument's sensitivity at dry conditions could be tested in the ozonolysis case showing that the instrument sensitivity drops by nearly an order of magnitude in the absence of water vapor. Because of the fast drop of the instrument's sensitivity for water vapor mixing ratios below 0.1 %, it is beneficial to add water vapor to the ion flow tube at very dry conditions of the sampled air to maintain a high instrument sensitivity.

Sanchez et al. (2016) used a similar approach to calibrate their instrument via photolysis of water, but they used water mixing ratios in the pptv range to keep $HO_2$ concentrations in an atmospheric range. They used purified air for the calibration source. This study uses synthetic air (purity 99.9999%).

Sanchez et al. (2016) found a constant sensitivity for water vapor mixing ratios between 0.2 and 0.8% whereas a 30% decrease is observed here. Only for one sensitivity measurement at 0.06% water mixing ratios an increased sensitivity by approximately 50% is reported by Sanchez et al. (2016). The reason for this different behaviour is not clear, but one may speculate that the design of the ion-flow tube and inlet nozzle might impact the collision probability of ion clusters. The relative change of the instrument's sensitivity in Sanchez et al. (2016) towards dry conditions is not reported, so that it is not clear, if the sensitivity drops for dry conditions in their instruments as observed here.

**3.3 Precision and uncertainty of the $HO_2$ measurement**

To determine the instrument's limit of detection, the Allan deviation was calculated from 2 hours of measurements, when no $HO_2$ was present. The signals of both masses at which $HO_2$ is detected (112 and 114) were taken into account for this analysis. The background signal was equivalent to $1 \times 10^8$ molecules cm$^{-3}$ and the count rate of the primary Br ion was $1 \times 10^5$ counts s$^{-1}$. The sensitivity during the measurement was $8 \times 10^{-12}$ cm$^3$ giving a count rate of 80 counts s$^{-1}$ (= 4800 counts min$^{-1}$) for the background signal. Poisson statistics predicts a noise that correlates to the square root of the counts, which fits well with the results of the Allan division plot shown in Fig. 4. This correlates to a signal with an expected noise of 70 counts that gives a 1-$\sigma$ limit of detection of $0.015 \times 10^8$ molecules cm$^{-3}$ for 60 s integration time. This is slightly better than the 1-$\sigma$ level of detection of $0.06 \times 10^8$ molecules cm$^{-3}$ reported for the instrument in Sanchez et al. (2016).

Uncertainties are caused by the calibration, which makes the major contribution of the measurement uncertainty with $\pm10\%$ (1$\sigma$) (Holland et al., 2003). The stability of the background signal in the measurements done here was $\pm12$ %, giving an upper

[Figure]

**Figure 4.** Allan deviation plot derived from sampling a constant $HO_2$ concentration of $1 \times 10^8\,HO_2\,molecules\,cm^{-3}$ over $5\,hours$. The Allan deviation demonstrates the precision of measurements depending on the integration time. The red line indicates the behavior of the Allan deviation, if the noise is only limited by Gaussian noise.

limit of the additional uncertainty from the stability of the subtracted background signal. Similar uncertainties are obtained by Sanchez et al. (2016).

**3.4 Instrumental background**

The instrumental background was characterized in experiments where the inlet was overflown with humidified synthetic air. This was done either using the radical source as a flow tube when the UV lamp was off or during experiments in SAPHIR, when only humidified synthetic air was present in the chamber. As shown in Fig. 5, the background signal changes similarly with water vapor for both experimental conditions. The shape of the water vapor dependence is consistent with the assumption that a constant $HO_2$ concentration $((1.5 \pm 0.2) \times 10^8\,molecules\,cm^{-3})$ is internally produced in the instrument, which is detected according to the water vapor dependence of the instrument sensitivity discussed above. Nevertheless, Fig. 5 shows that the background was up to 20% lower in the laboratory measurement and the measured background shows a better linearity compared to the chamber measurements. Both backgrounds were calibrated using a water dependent calibration.

The background can be subtracted from the measured $HO_2$ concentration after applying the water vapor dependent calibration factor. The value of the background needs to be regularly determined. As reported for other CIMS instruments detecting radicals (Berresheim et al. (2000); Sanchez et al. (2016)), the radicals can be produced by the ion source. Therefore, this is the likely reason for the observed background signal. For chamber experiments reported here, the background signal was measured in the clean dark chamber at the start of each experiment. No trend of the background signal over a period of 2 month was

[Figure]

**Figure 5.** The background $HO_2$ measurement in the SAPHIR chamber derived during the humidification of the clean chamber and the background measured during the laboratory calibration supplying humidified synthetic air.

observed. The day-to-day variability of the background (in total 16 experiments) was within a range of $\pm 12\,\%$ during 2 months of measurements at the chamber.

Sanchez et al. (2016) also described a constant $HO_2$ source which causes a background. An $HO_2$ titration experiment Sanchez et al. (2016) confirmed that $HO_2$ is internally produced, which has been discussed for other radical measurements us-
5  ing a CIMS approach (Berresheim et al., 2000). Sanchez et al. (2016) determined an instrument background of at least 4 pptv $HO_2$, which compares well with the background of 6 pptv $HO_2$ that has been found during the experiments in the SAPHIR chamber.

**3.4.1 Potential interference from ozone**

Ozone is known to be an interference in some $HO_2$ LIF instruments due to the photolysis of $O_3$ by the 308 nm excitation laser
10  (Holland et al., 2003). In order to test whether ozone can also cause an interference in the CIMS detection of $HO_2$, laboratory experiments were performed. Ozone was added to humidified synthetic air (water vapor mixing ratios 0.3 and 2.6 %). For both conditions no increase of the CIMS background signal could be observed for ozone mixing ratios of up to 400 ppbv. Details of the experiment are shown in the supplementary material. Results are consistent with the laboratory characterization experiments performed by Sanchez et al. (2016) for their Br- CIMS instrument.
15     During experiments in the SAPHIR chamber, instrument background effects can only be determined for periods of the experiments without the presence of reactants, when no $HO_2$ was present. A time series for a typical experiment is shown in

Fig. 6. Typically, ozone was added in a concentration of 100 to 200 ppbv. Although no artefacts were found in the laboratory characterization, an increase in the background upon ozone addition was observed in two of 12 experiments in SAPHIR. For these two experiments, the chamber was first humidified and ozone was added afterwards. The increased background appears as an increased intercept of $2.3 \times 10^8$ and $1.0 \times 10^8$ HO$_2$ molecules cm$^{-3}$ in the linear regression between LIF and CIMS HO$_2$

5    data for the experiments of 21 June and 26 June (Fig. 7), respectively. The data of the LIF instrument were corrected for a maximum ozone interference of $0.05 \times 10^8$ and $0.15 \times 10^8$ HO$_2$ molecules cm$^{-3}$ on these days, respectively. This correction is much smaller than the HO$_2$ concentration observed by the CIMS instrument, so that it can be excluded that differences are due to systematic errors in the data of the LIF instrument.

      In the correlation plot (Fig. 8), including all experiments, this additional background was subtracted. The increased back-

10   ground due to the ozone addition will be investigated in further chamber experiments. Because no direct connection between the occurrence of this interference and chemical conditions in the experiments is observed, it might be related to instrumental effects that could vary with time such as cleanness of the ion flow tube walls. ==This indicates that regular checks of the background signal is needed to take an appropriate background correction into account.==

**3.5   Comparison of CIMS and LIF HO$_2$ measurements**

[Figure]

**Figure 6.** Time series plot for the HO$_2$ concentrations measured by the CIMS and the LIF instrument during the photo-oxidation experiment at 19 June 2017 in the SAPHIR chamber. The gray shaded area indicates that the chamber roof was closed. The vertical lines are showing the injection time of additional reactants, in case of water the injection took longer indicated by a broader line.

[Figure]

**Figure 7.** Correlation between $HO_2$ measurements by the CIMS and LIF instruments for individual chamber experiments. Labels in the plots indicate the specific VOC injected into the chamber. For the regression line shown in blue a least square fit was performed.

The $HO_2$ production was initiated with the injection of ozone and the opening of the chamber roof providing UV light to the chamber, as shown in the time series in Fig. 6. An addition of CO further boosted the $HO_2$ production, which dropped upon closing of the roof. After the injection of water the CIMS shows a stable signal with a small offset. During the experiment the LIF and CIMS data reveal a good correlation. This experiment was performed without the addition of a volatile organic compound (VOC), as well as, two other experiments marked with "None" in Fig. 7. Nevertheless, $HO_2$ is produced in these experiments, because OH and NO are produced from the photolysis of HONO released from the Teflon chamber walls in the sunlit chamber (Rohrer et al., 2005). Reaction of small concentrations of OH reactants formed under these conditions in the chamber lead to the formation of $HO_2$ (Rohrer et al., 2005).

Figure 7 displays the correlation between $HO_2$ measurements by the CIMS and the LIF instrument for all day-long photo-oxidation experiment in the SAPHIR chamber performed in this study. The results of a linear regression analysis are given in Fig. 7, which takes errors in both $HO_2$ measurements into account (Press et al., 1992). The chemical composition was varied between experiments by changing for example the NO mixing ratio. The different chemical conditions during the experiments

allows for checking for potential interferences. High NO concentrations of up to 3 ppbv were reached by injecting NO to the chamber air on 31 May and 02 June, and up to 80 ppbv $NO_2$ was added on 23 June. The $NO_2$ interference test was performed by injecting $NO_2$ in the dark, dry chamber. No further photo-chemistry experiments was done on this particular day. No systematic change in the relation between $HO_2$ related to the presence of NO or $NO_2$ from the two instruments is

5   observed in these cases (Fig. 7), which is in agreement with the results of Sanchez et al. (2016). In general, no interference from VOCs (Isoprene, ISOPOOH and reaction products) are observed, except for experiments with IEPOX injections. IEPOX was detected on m/z 197 as $Br^- \cdot IEPOX$ ion cluster, but the instrument was not calibrated for IEPOX. Nevertheless, this mass trace can be used to correct the $HO_2$ measurement for the interference from IEPOX, the correction is shown in the supplementary material. The correlation plots shown in Fig. 7 are corrected for the IEPOX interference. The $HO_2$ signal

10   observed during the injection of IEPOX can be attributed to the interference from IEPOX, because IEPOX was injected in the dark chamber so that no $HO_2$ is expected to be present. This gives the relationship between the signal observed at the IEPOX mass (m/z 197) to the interference signal from IEPOX at the $HO_2$ mass (m/z 112). During the photo-oxidation of IEPOX, when also $HO_2$ is present, the interference signal can be subtracted from the signal at the $HO_2$ mass by scaling the initial interference signal by the relative change on m/z 197. The correction improves the correlation of the CIMS and the LIF but the

15   absolute agreement is still not as good (slope of the regression 0.93; coefficient of determination 0.79) compared to the other experiments. The corrections are in the order of or smaller than the $HO_2$ measurements, and works best for the experiment with the lowest IEPOX concentration. A plausible reason for the IEPOX interference found seems to be a fragmentation of the cluster ion in the transfer stage of the instrument. The fragmentation could be initiated by acceleration of the ions in the electrostatic field causing collisions with other molecules. It is worth noting that IEPOX concentrations were at least 10 times

20   higher than typically found in the atmosphere. Kaiser et al. (2016) found IEPOX concentrations of 1 ppbv during a campaign in a forest in the South-East US where isoprene, the precursor of IEPOX, was the dominant organic species. Therefore no significant interference for atmospheric measurements by the CIMS instrument are expected from IEPOX.

During experiments with ISOPOOH, $HO_2$ measurements by the LIF instrument showed higher values than $HO_2$ measured by the CIMS instrument (slope of the linear regression of 0.88; coefficient of correlation $R^2 = 0.68$). Further experiments will

25   be needed to investigate if ISOPOOH could cause an interference in the LIF instrument. Like in the case of IEPOX, ISOPOOH concentrations were much higher (several ppbv) than typically found in the atmosphere (less than 1 ppbv Kaiser et al. (2016)), so that no significant impact for atmospheric conditions is expected.

All concurrent measurements of the two instruments for $HO_2$ by CIMS and LIF, in the photo-oxidation experiments are summarized in the correlation plot shown in Fig. 8. In general, the correlation fit shows that there is an excellent agreement

30   of both instruments giving a slope of linear regression of 1.14 and the linear correlation coefficient $R^2$ is 0.87. Experiments investigating the photo-oxidation of IEPOX and ISOPOOH are color-coded and are excluded from the correlation fit. However, using all data for the correlation fit leads to similar result (slope of linear regression of 0.86; coefficient of correlation $R^2 = 0.89$).

Correlation of individual experiments (Fig. 7, e.g. 21 June and 26 June) give partly significant offsets in the regression

35   analysis of up to $2.3 \times 10^8 \, cm^{-3}$ $HO_2$. One possible reason could be the procedure, how the water vapor dependence of the

[Figure]

**Figure 8.** Correlation plot for the $HO_2$ concentrations measured by the CIMS and the LIF instrument of all photo-oxidation experiments in the SAPHIR chamber. A linear fit is applied to the subset of data excluding experiments with IEPOX and ISOPOOH.

instrument sensitivity was derived. This was done by using the measured signal at the $HO_2 \cdot Br^-$ mass during the humidification process of the clean chamber air, when no $HO_2$ was present. However, the chamber air might not be perfectly mixed during the humidification, because water vapor from boiling water is introduced at one location in the chamber together with a high flow of synthetic air. Because the water measurement in the chamber used for the determination of the CIMS background

5 signal and the CIMS inlet are at different locations in the chamber, the water measurement is potentially not accurate for the water vapor sampled by the CIMS for these conditions, so that small systematic errors in the background determination cannot be excluded. In the future, the water vapor dependence of the background will be determined independently from the chamber experiment, so that it can be expected that such effects will not be relevant.

**4 Conclusion and Outlook**

10 Chemical ionization was applied to measure atmospheric $HO_2$ concentrations using bromide ions as reagent. Laboratory characterization experiments and measurements in the atmospheric simulation chamber SAPHIR in Jülich were used to check the instruments applicability for atmospheric measurements.

The performance of the CIMS instrument is comparable with measurements by a laser-induced fluorescence instrument. A water vapor dependence of the instrument sensitivity needs to be taken into account in the evaluation of data because the

15 sensitivity of the instrument changes by roughly a factor of 2 for atmospheric water vapor concentrations between 0.2 and 1.4 %. Also a water vapor dependent background signal is observed. The change of the background signal with increasing

water vapor, however, is explained by the water vapor dependence of the sensitivity. Therefore, the assumption is that the background consists of constant $HO_2$ production in the instrument. This background was stable within $\pm 12\%$ during two months of measurements and no further trend was identified. The background signal and the instrument detection sensitivity needs to be quantified on a daily basis.

5     No significant interference from trace gases NO, $NO_2$, $O_3$, CO, isoprene and ISOPOOH were found for atmospheric conditions. Only for non-atmospheric high IEPOX concentrations of several ppbv artificial signals were found that scaled with the IEPOX concentration. The $HO_2$ measurements correlate well with the LIF measurements. A slope of the linear regression of 1.07 was determined and a linear correlation coefficient ($R^2$) of 0.87 was found.

     $HO_2$ was directly sampled through a nozzle into a custom-build ion flow tube which was optimized for sensitivity. The
10   sensitivity reached is equal to $0.005 \times 10^8$ $HO_2$ per $cm^3$ for $10^6$ cps of bromide and 60 s of integration time, which is approximately 3 times higher than the sensitivity for a similar instrument by Sanchez et al. (2016). Therefore the instrument is suitable to measure typical $HO_2$ concentrations in the atmosphere. Further the Allan deviation shows that the instrument follows Gaussian noise allowing integration time of up to 500 s.

     Chemical conditions in the chamber experiments were close to atmospheric conditions regarding the most important con-
15   stituents of the atmosphere such as $NO_x$, ozone and water vapor showing the applicability of the instrument under these conditions. First future deployment in field experiments will be done with concurrent $HO_2$ measurements by the LIF instrument, so that potential so far unrecognized interference can be identified.

     For future application of the instrument in field and chamber experiments, various modifications of the instrument will be tested to improve the sensitivity and minimized the background signal: A sheath flow of pure nitrogen in the ion flow tube could
20   help to prevent wall contact of radicals in the ion-flow tube. Further, the sheath flow could be humidified to prevent sensitivity loss for measurements performed in dry conditions. Additionally, an automated calibration will be installed to perform daily calibration and background measurements. An important benefit of the instrument is that the bromide ion chemistry can also detect organic compounds specifically oxygenated organic compounds and acids. Therefore the technique provides a valuable tool in future field and simulation experiments.

[revised manuscript text omitted]